# Structural insights into catalytic mechanism and product delivery of cyanobacterial acyl-acyl carrier protein reductase

Yu Gao[1,5], Hongmei Zhang [1,5], Minrui Fan[1,3], Chenjun Jia[1,4], Lifang Shi[1], Xiaowei Pan [1], Peng Cao[1], Xuelin Zhao[1], Wenrui Chang[1,2] & Mei Li [1✉]

Long-chain alk(a/e)nes represent the major constituents of conventional transportation fuels. Biosynthesis of alkanes is ubiquitous in many kinds of organisms. Cyanobacteria possess two enzymes, acyl-acyl carrier protein (acyl-ACP) reductase (AAR) and aldehyde-deformylating oxygenase (ADO), which function in a two-step alkane biosynthesis pathway. These two enzymes act in series and possibly form a complex that efficiently converts long chain fatty acyl-ACP/fatty acyl-CoA into hydrocarbon. While the structure of ADO has been previously described, structures of both AAR and AAR–ADO complex have not been solved, preventing deeper understanding of this pathway. Here, we report a ligand-free AAR structure, and three AAR–ADO complex structures in which AARs bind various ligands. Our results reveal the binding pattern of AAR with its substrate/cofactor, and suggest a potential aldehyde-transferring channel from AAR to ADO. Based on our structural and biochemical data, we proposed a model for the complete catalytic cycle of AAR.

[1] National Laboratory of Biomacromolecules, CAS Center for Excellence in Biomacromolecules, Institute of Biophysics, Chinese Academy of Sciences, Beijing 100101, People's Republic of China. [2] University of Chinese Academy of Sciences, Beijing 100049, People's Republic of China. [3] Present address: Department of Molecular and Cellular Physiology, 279 Campus Drive, Stanford University School of Medicine, Stanford, CA 94305, USA. [4] Present address: CAS Key Laboratory of Receptor Research, Shanghai Institute of Materia Medica, Chinese Academy of Sciences, Shanghai 201203, People's Republic of China. [5] These authors contributed equally: Yu Gao, Hongmei Zhang. ✉email: meili@ibp.ac.cn

The demands for petrol-based fuels has remained high, calling for alternative fuel sources. Biofuels are of particular interest as a renewable supplement for gradually diminishing fossil fuels[1–4]. Long-chain alk(a/e)nes are the major constituents of conventional transportation fuels. Biosynthesis of alkanes represents a promising way to produce substitutes for petroleum-based fuels[5–8]. Long-chain alk(a/e)nes are commonly produced from fatty acid metabolites in various species[4,9–13]. The earlier discovery of the pathway responsible for alkane biosynthesis in cyanobacteria allowed the conversion from solar energy to alkanes through photosynthesis, without providing external carbon sources[14]. This two-step pathway utilizes two water-soluble enzymes, namely acyl-acyl carrier protein (acyl-ACP) reductase (AAR) and aldehyde-deformylating oxygenase (ADO). AAR catalyzes the first step of this pathway, reducing the substrate fatty acyl-ACP/fatty acyl-coenzyme A (acyl-CoA) into the corresponding aldehyde in a NADPH-dependent manner[14,15]. Subsequently, ADO catalyzes the conversion of fatty aldehyde into alk(a/e)ne with one carbon shorter ($C_{n-1}$ hydrocarbon) and the formate as a $C_1$-derived form[14,16].

However, this pathway yields little alkane products, primarily because of the slow turnover of the two enzymes[15–18]. Thus, structure-based engineering of the two enzymes is a practical way to optimize the efficiency of alkane synthesis. Several crystal structures of ADO from different species of cyanobacteria were previously solved, revealing that ADO adopts a ferritin-like eight-helix (henceforth named helix 1–8) architecture, with a di-iron center buried inside the molecule[19–22]. An occluded substrate binding cavity accessing the di-iron center was identified in these structures[19,21,22]. In addition, a hydrophobic tunnel featuring an opening between helix 7 and helix 8 was suggested to be the substrate entering tunnel based on the structures of ADO from *Prochlorococcus marinus* (*Pm*ADO). Together, the substrate cavity and the entering tunnel form a "T-shaped" channel inside ADO. ADO is considered to be the rate-limiting enzyme in this pathway[18]. Based on the available structural information, a number of ADO mutants were generated and shown to possess higher activity compared with the wild type[23].

The catalytic reaction of AAR was shown to occur in two steps. Firstly, the substrate (fatty acyl-ACP or fatty acyl-CoA) was found to be associated with AAR, with the fatty acyl moiety covalently bound with the residue C294 of AAR (ref. [18]). As a result, an acyl-enzyme intermediate is formed, and the ACP or CoA moieties are released. Subsequently, a corresponding aldehyde is produced from the acyl-enzyme intermediate in the presence of NADPH (Supplementary Fig. 1). AAR was previously shown to prefer stearoyl-CoA among acyl-CoA esters of different length and saturation[15], and subsequently it was shown to exhibit the highest activity toward stearoyl-ACP (ref. [18]). Previously reported results suggested that AAR utilizes a "Ping-pong" mechanism, binding fatty acyl substrate prior to binding the cofactor NADPH during the catalysis[15,18]. To date, no structure of either AAR alone or AAR in complex with substrate or NADPH has been solved. In addition, due to the poor solubility of long chain aldehydes, it had remained unclear how are the aldehydes transferred between two water-soluble enzymes in the hydrophilic environment[15,16,18,24]. A recent report suggested that together, AAR and ADO forms a tight complex that, by facilitating the aldehyde transferring from AAR to ADO either in a direct or an indirect way, increases the efficiency of the alkane biosynthesis[18]. However, due to the current lack of the complex structure, the precise details of the pathway of aldehyde transferring from AAR to ADO have remained undetermined.

Here we report the crystal structure of AAR from *Synechococcus elongatus PCC* 7942 (*Se*AAR) in its apo form (AAR$_{apo}$), together with three structures of AAR in complex with ADO from the same species of cyanobacteria (*Se*ADO), in which AAR binds either substrate or substrate/cofactor. Our results provide details about the binding mode of AAR with ligands, the interaction between AAR and ADO, as well as the potential aldehyde-transferring pathway between the two enzymes. AAR in the four structures likely represents different state during its catalytic process, and our results shed light on the catalytic mechanism of AAR.

## Results

**The AAR$_{apo}$ structure represents the resting state of AAR.** In this work, we solved the AAR$_{apo}$ structure at 2.8 Å resolution (Table 1), and modeled the complete AAR protein containing 341 amino acid residues. As shown in Fig. 1a, AAR is composed of three domains: the N-terminal domain (NTD, residues 1–130), the middle domain (mid-domain, residues 131–264), and the C-terminal domain (CTD, residues 265–341). A conserved loop, $^{162}$GATGDIG$^{168}$, in the mid-domain was designated as the dinucleotide recognition loop[25], and contains the classical sequence motif $GX_{(1-3)}GX_{(1-2)}G$ for binding NAD(P)$^+$ or NAD(P)H (Supplementary Fig. 2). Residue C294, which is strictly conserved in AAR homologues and has been identified as the key residue for catalysis[18], is located approximately at the center of AAR molecule. Above C294, a wide cleft that includes the dinucleotide recognition loop is visible (Fig. 1a). This structural observation suggests that the cleft constitutes the binding site for NADPH and the entrance for substrate access to the catalytic residue C294.

In the AAR$_{apo}$ structure, we built five molecules, which show highly similar overall architectures, in an asymmetric unit. However, three fragments, namely Fr-I (residues R19–Q29) and Fr-II (S113–T120) from the NTD, and Fr-III (V221–D235) from the mid-domain, show higher structural diversity (Supplementary Fig. 3a), indicating that these regions possess an intrinsic flexibility. In fact, we found that the mobility of these fragments is pivotal for the function of AAR, as they are involved in either the interaction with ADO (Fr-I and Fr-II), or NADPH binding (Fr-III), as discussed later.

**The AAR–ADO complex and the intermolecular interactions.** We determined three ternary complex structures of AAR–ligand bound with ADO, which are similar in their overall architecture (Supplementary Fig. 3b). We named these structures based on which ligands associated with AAR, i.e., AAR$_{ligand}$–ADO (Table 1). In these complex structures, ADO exhibits a nearly identical conformation to that observed in the previously reported structures of *Se*ADO alone[21]. ADO adopts an eight-helix folding, and contains two iron atoms and a hydrocarbon chain bound at the substrate cavity (Supplementary Fig. 3b). While helix 7 of ADO interacts with the NTD of AAR (Fig. 1b, c), the acidic residues of this helix form strong electrostatic interactions with the basic residues from the long helical region (R73–H91), as well as R118 from Fr-II of AAR (Fig. 1c, Supplementary Fig. 3c). These residues are highly conserved in homologues of AAR (Supplementary Fig. 2) and ADO, highlighting their crucial roles in forming the complex.

To further verify the AAR–ADO interface observed in the complex structures, we mutated each of the five acidic residues from helix 7 of ADO, and further generated a double mutant E200A/D201A and triple mutant E196A/E200A/D201A (Supplementary Fig. 4). We then measured the binding affinities of AAR with the wild type and the seven mutant forms of ADO using isothermal titration calorimetry (ITC; Table 2, Supplementary Fig. 5). The binding assays revealed that the wild-type forms of AAR and ADO formed a complex with a measured $K_D$ value of

**Table 1 Summary of data collection and structure refinement.**

| | AAR-His (Hg-derivative) | AAR$_{apo}$ | AAR$_{thioester}$–ADO | AAR$_{NADPH}$–ADO | AAR$_{stearoyl-CoA}$–ADO |
|---|---|---|---|---|---|
| PDB code | | 6JZQ | 6JZU | 6JZY | 6JZZ |
| Data collection | | | | | |
| Space group | C2 | C2 | P2$_1$2$_1$2$_1$ | P2$_1$2$_1$2$_1$ | P2$_1$2$_1$2$_1$ |
| Cell dimensions | | | | | |
| $a, b, c$ (Å) | 56.7, 180.9, 90.3 | 92.2, 201.4, 124.6 | 66.4, 71.3, 137.1 | 65.6, 71.3, 135.7 | 66.5, 72.3, 137.2 |
| $\alpha, \beta, \gamma$ (°) | 90, 90.03, 90 | 90, 105.9, 90 | 90, 90, 90 | 90, 90, 90 | 90, 90, 90 |
| Resolution (Å) | 50.0 − 2.30 (2.38 − 2.30) | 46.11 − 2.80 (2.90 -2.80) | 48.59 − 2.18 (2.26 − 2.18) | 48.26 − 2.10 (2.17 − 2.10) | 38.66 − 3.01 (3.12 − 3.01) |
| $R_{merge}$ | 0.08 (0.49) | 0.086 (0.80) | 0.086 (1.08) | 0.08 (0.99) | 0.089 (>1) |
| $I / \sigma I$ | 22.6 (4.2) | 23.6 (2.8) | 30 (2.8) | 32 (2.4) | 27.4 (2.5) |
| Completeness (%) | 99.9 (99.9) | 99.5 (99.2) | 99.6 (99.3) | 99.1 (99.0) | 99.9 (100) |
| Redundancy | 6.8 (6.9) | 7.6 (7.7) | 13.1 (13.1) | 12.7 (12.8) | 14.0 (14.4) |
| Refinement | | | | | |
| Resolution (Å) | 50 − 2.30 (2.38 − 2.30) | 46.11 − 2.80 (2.90 − 2.80) | 49.43 − 2.18 (2.26 − 2.18) | 49.13 − 2.10 (2.17 − 2.10) | 38.66 − 3.01 (3.12 − 3.01) |
| No. reflections | | 53,258 (5159) | 34,321 (3189) | 37,568 (3434) | 12,814 (721) |
| $R_{work}/R_{free}$ | | 0.22/0.25 | 0.20/0.25 | 0.21/0.24 | 0.21/0.27 |
| No. atoms | | | | | |
| Protein | | 12948 | 4338 | 4346 | 4334 |
| Ligand/ion | | | 38 | 86 | 75 |
| Water | | 49 | 163 | 143 | 16 |
| B-factors | | | | | |
| Protein | | 74.23 | 36.41 | 57.93 | 50.30 |
| Ligand/ion | | | 39.39 | 55.02 | 47.74 |
| Water | | 57.96 | 36.47 | 52.45 | 36.99 |
| R.m.s. deviations | | | | | |
| Bond lengths (Å) | | 0.012 | 0.008 | 0.008 | 0.012 |
| Bond angles (°) | | 1.45 | 0.89 | 0.89 | 1.36 |

Values in parentheses are for the highest-resolution shell.

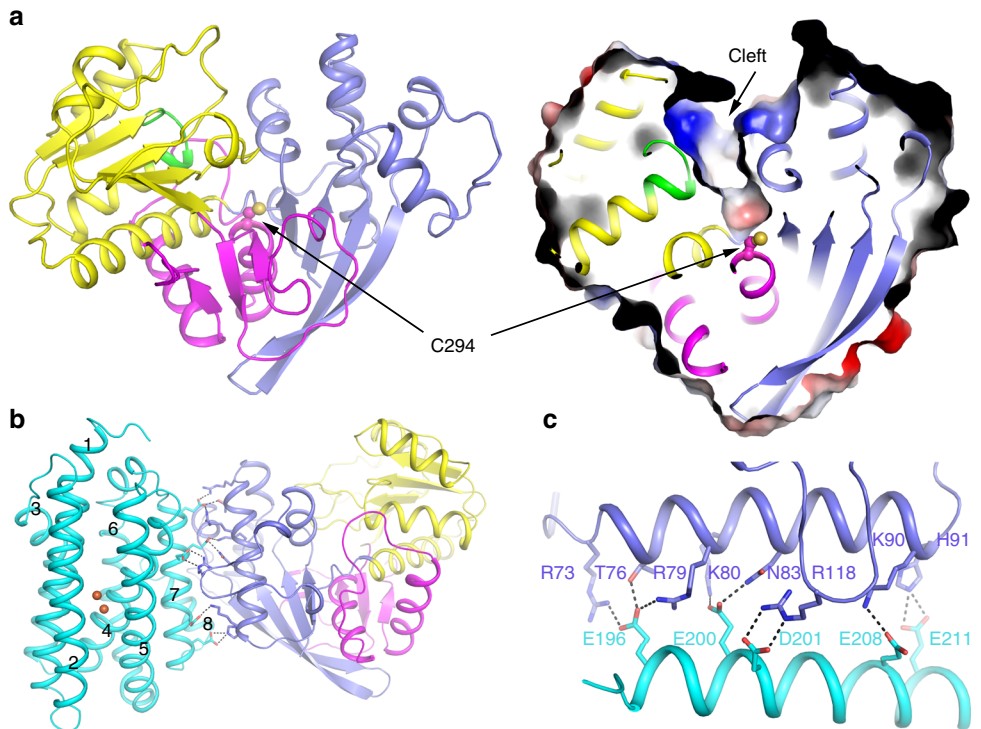

**Fig. 1 Structures of AAR and AAR–ADO complex. a** Cartoon (left) and surface (right) representation of AAR$_{apo}$ structure. NTD, mid-domain, and CTD of AAR are shown in slate, yellow and magenta, respectively. The dinucleotide recognition loop in the mid-domain is highlighted in green. The key residue C294 is shown in stick-ball mode and labeled. The cleft above the residue C294 was indicated by an arrow in the right panel. **b** Cartoon representation of AAR–ADO complex structure (shown by AAR$_{NADPH}$–ADO structure). NTD, mid-domain, and CTD of AAR are shown in slate, yellow and magenta, respectively. ADO is shown in cyan and the eight helices are labeled as 1–8. The two iron atoms in ADO are shown in orange spheres. The residues involved in the intermolecular interaction are shown in sticks. The salt bridges and hydrogen bonds are shown by black dashed lines. The bound ligands are omitted for clarity. **c** Details of the intermolecular interaction between AAR and ADO. For clarity, only fragments of AAR (R73–H91 and L114–L122) and ADO (helix 7) are shown. The residues involved in the intermolecular interactions are shown in sticks and labeled.

2.0~2.2 μM, which is comparable to that of the similar AAR–ADO complex from *Nostoc punctiforme* (*Np*)[18]. The mutants E196A and E200A exhibit significantly reduced binding affinity with AAR, whereas the E200A/D201A and E196A/E200A/D201A mutations completely abolished the capability of ADO to bind with AAR (Table 2, Supplementary Fig. 5). We further mutated the basic residues of AAR involved in the interactions with E196 and E200, creating three AAR single mutants, R73A, R79A, and K80A (Supplementary Fig. 4). The binding assay between AAR (wild type and mutants) and ADO

confirmed the essential role of the R73(AAR)–E196(ADO), R79 (AAR)–E196(ADO), and K80(AAR)–E200(ADO) pairs for the AAR–ADO complex formation (Table 2, Supplementary Fig. 6). Together, our results demonstrate that the AAR–ADO complex observed in our structure is representative of the complex that exists in physiological conditions.

**The AAR_thioester–ADO structure.** During catalysis, AAR first binds its substrate acyl-CoA or acyl-ACP to form an acyl-enzyme intermediate with a covalent thioester bond between the acyl moiety and C294 of AAR. In our AAR_thioester–ADO structure solved at 2.18 Å resolution, we observed a long tube-shaped electron density, which can be well fitted with a stearic chain, within an interior tunnel of AAR (Fig. 2a, b, Supplementary Fig. 7a). No electron density corresponding to the CoA moiety was observed, despite the incubation of stearoyl-CoA with AAR prior to crystallization. This finding indicates that the acyl-enzyme intermediate was spontaneously formed, and the CoA moiety had already been released. The structure showed that the C1 atom of the stearoyl chain (stearoyl-C1) had formed a covalent thioester bond with the hydrosulfide group of C294 (C294-SH; Fig. 2b, Supplementary Fig. 7a). Mass spectrometric analysis of the stearoyl-CoA-incubated AAR sample that was used for crystallization confirmed the thioester bond formation (Supplementary Fig. 7b), thus the AAR_thioester–ADO structure represents the acyl-enzyme intermediate state of AAR. In the AAR_thioester–ADO structure, the stearic chain is buried inside a hydrophobic tunnel (acyl-tunnel) formed mainly by non-polar residues from the NTD and CTD of AAR (Fig. 2c). One end of the acyl-tunnel connects with the wide cleft above C294, forming an L-shaped, long channel (Fig. 2b). At the other end, the acyl-tunnel can be seen to extend toward the AAR–ADO interface with its opening

**Table 2 Binding affinity between AAR and ADO measured through ITC method.**

| | $K_D$ value (μM) |
|---|---|
| **Binding with AAR** | |
| ADO wild type | 2.0 ± 0.50 |
| ADO E196A | N.D. |
| ADO E200A | N.D. |
| ADO D201A | 4.9 ± 1.40 |
| ADO E208A | 14.8 ± 3.70 |
| ADO E211A | 2.8 ± 0.85 |
| ADO E200A/D201A | N.D. |
| ADO E196A/E200A/D201A | N.D. |
| **Binding with ADO** | |
| AAR wild type | 2.2 ± 0.30 |
| AAR Y247F | 2.2 ± 0.40 |
| AAR R73A | N.D. |
| AAR R79A | N.D. |
| AAR K80A | N.D. |
| AAR Y247A | N.D. |

*N.D. not detected.*

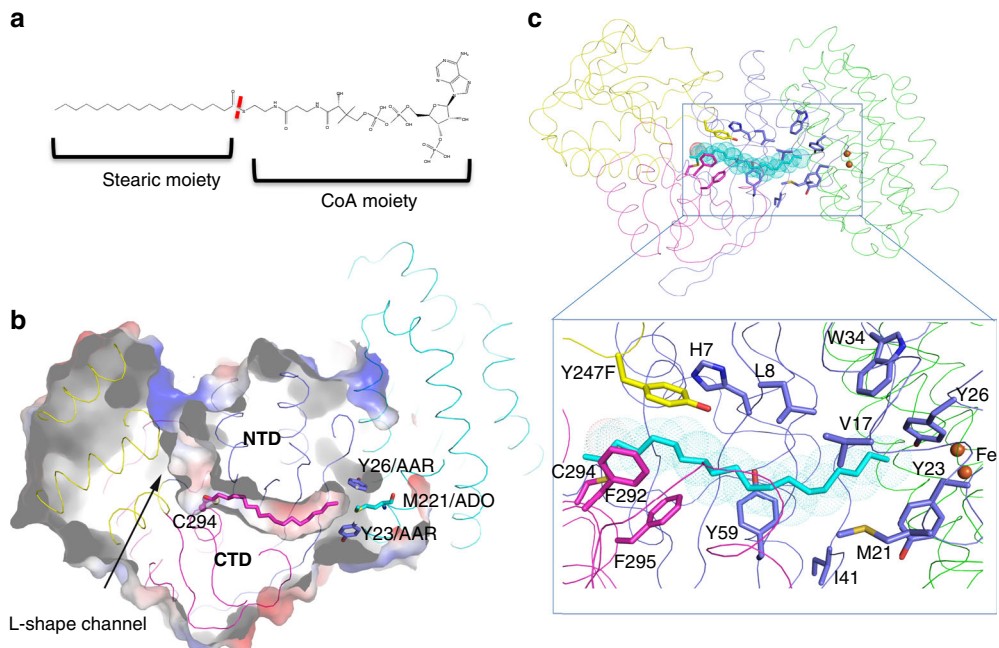

**Fig. 2 AAR_thioester–ADO structure. a** Chemical structure of the substrate stearoyl-CoA. The red dashed line indicates the thioester bond between stearic chain and CoA moiety. **b** Surface representation of AAR_thioester–ADO structure. The L-shaped channel within AAR is indicated. The stearic chain (shown as magenta stick), covalently bound with C294 (shown in stick-ball mode), is located in the hydrophobic tunnel, which is a part of the L-shaped channel. Residues Y23 and Y26 from AAR, and M221 from ADO, forming the terminal end of the tunnel, are shown in stick and labeled. The color code of AAR and ADO is the same as that in Fig. 1b. **c** The hydrophobic residues, mainly from the NTD and the CTD, are involved in the substrate tunnel formation and shown as sticks. The color code of AAR is the same as that in Fig. 1b, and ADO is shown in green.

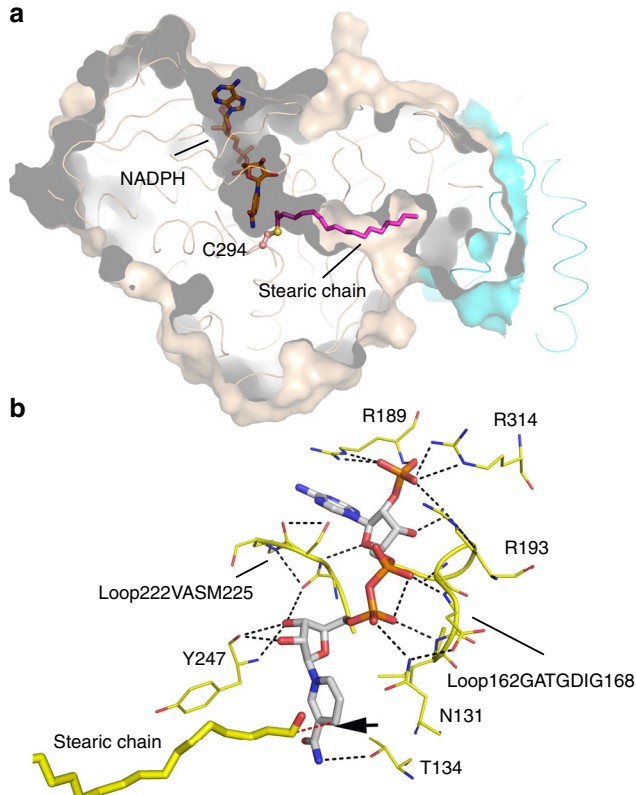

**Fig. 3 AAR$_{NADPH}$–ADO structure. a** Surface representation of AAR$_{NADPH}$–ADO structure. AAR and ADO are shown in wheat and cyan, respectively. NADPH and stearic chain (shown as stick) are accommodated in the L-shaped channel. The residue C294 is highlighted in stick-ball mode. **b** The interaction of NADPH with adjacent residues of AAR. Two loops of AAR sandwiching NADPH are shown in ribbon mode. NADPH and the stearic chain are colored white and yellow, respectively, for the carbon atoms and shown in sticks. Residues involved in the NADPH interactions are shown in lines. The hydrogen bonds are shown by black dashed lines. The C4 atom of NADPH (indicated by black arrow) is close to the stearoyl-C1 atom with a distance of 3.14 Å (red dashed line).

surrounded by residues Y23 and Y26 from the NTD of AAR and sealed by M221 of ADO (Fig. 2b). In the AAR$_{apo}$ structure, this acyl-tunnel is broken by the side chains of M279 and M21 (Supplementary Fig. 7c). Both M279 and M21 undergo significant conformational changes in the AAR$_{thioester}$–ADO structure, where they participate in forming the acyl-tunnel. Together, these findings indicate that substrate binding induces formation of the acyl-tunnel in AAR, as a result of that the stearic chain is well accommodated in the tunnel.

**The AAR$_{NADPH}$–ADO structure**. The molecule NADPH is an essential cofactor for catalysis, as it provides a hydride to the thioester intermediate, enabling aldehyde production[18,25]. In order to identify the NADPH binding pattern, we either co-crystallized AAR or soaked the AAR–ADO crystals with NADPH, and finally solved the AAR$_{NADPH}$–ADO structure at 2.1 Å resolution by soaking the AAR$_{thioester}$–ADO crystal with NADPH. In the structure, one NADP(H) molecule is accommodated in the hydrophilic cleft above C294 (Fig. 3a, Supplementary Fig. 8), and is mainly stabilized by the mid-domain of AAR. The ADP moiety is sandwiched by the dinucleotide recognition loop and the $^{222}$VASM$^{225}$ loop region within Fr-III (Fig. 3b). The 3′-phosphate group of the ADP moiety is located in

### Table 3 Binding affinity and enzymatic activity of the wild type and mutants of AAR toward stearoyl-CoA.

| Protein | Binding assay | Activity assay |
|---|---|---|
| | $K_D$ value (μM)[a] | Octadecanal yield (μg mL$^{-1}$) |
| AAR-wt (−K$^+$)[b] | 2.3 ± 0.38 | 6.0 ± 0.76 |
| AAR-wt (+K$^+$)[b] | 0.1 ± 0.05 | 110.9 ± 41.25 |
| AAR-Y247F (+K$^+$)[b] | 0.5 ± 0.15 | 61.1 ± 6.76 |
| AAR-Y247A (+K$^+$)[b] | N.D.[c] | – |

[a]The binding affinity of wild type (wt) or mutants of AAR with the substrate stearoyl-CoA measured by ITC.
[b](−K$^+$) and (+K$^+$) mean that the assays were measured in the absence and presence of potassium ion, respectively.
[c]N.D. not detected.
Source data are provided as a Source Data file.

a pocket surrounded by three positively charged residues: R189, R193, and R314 (Fig. 3b). These three residues are highly conserved among AAR homologues (Supplementary Fig. 1), and may account for the previously reported NADPH-dependent (instead of NADH-dependent) activity of AAR (ref. [15]).

In the AAR$_{NADPH}$–ADO structure, the nicotinamide moiety of NADPH is located close to C294 of AAR, with its C4 atom being located only 3.14 Å away from the stearoyl-C1 atom (Fig. 3b). The short distance supports the possibility that C4 of NADPH provides a hydride to break the thioester bond. The stearic chain is located in the acyl-tunnel of AAR, as was observed in the AAR$_{thioester}$–ADO structure. However, the weaker electron density of the stearic chain in the AAR$_{NADPH}$–ADO structure suggests that it only partially occupies the acyl-tunnel. Previous studies reported that AAR requires divalent metal ions (Mg$^{2+}$) for catalysis, and that its activity is stimulated ~10–20-fold by monovalent metal ions, including K$^+$ ions[15,18]. In the present work, ions such as Mg$^{2+}$ and K$^+$ were always provided in the purification and crystallization buffers of AAR. The introduction of NADPH should enable AAR to proceed with catalysis in the presence of cation ions. This process may have occurred slowly within the crystals, as a result of that aldehyde was produced and released in a proportion of the AAR molecules, while the thioester bond was preserved in other AAR molecules in the crystals. Together, these results suggest that the AAR$_{NADPH}$–ADO structure represents the catalytic state of AAR.

Previous results[15,18] together with our enzymatic assay (Table 3) all demonstrated that the presence of K$^+$ ions results in greatly increased activity of AAR. These results implied that AAR directly binds K$^+$ ions. To verify this assumption, we measured the binding affinity of K$^+$ ions with AAR, using ITC method. Under our experimental conditions, we failed to detect signs of AAR binding K$^+$ ions; however, we found that the binding affinity of AAR to its substrate stearoyl-CoA was significantly higher in the presence of K$^+$ (Table 3, Supplementary Fig. 9), suggesting that monovalent ions assist substrate binding and thus stimulate the catalytic activity of AAR. It is possible that the K$^+$ ions enhance the solubility of stearoyl-CoA, as previously suggested for palmitoyl-CoA[26].

**The AAR$_{stearoyl-CoA}$–ADO structure**. We obtained the AAR$_{stearoyl-CoA}$–ADO structure using the AAR C294S mutant, which avoid spontaneous formation of the acyl-enzyme intermediate, and solved the structure at 3.01 Å resolution. In this structure, the entire stearoyl-CoA molecule occupies the long L-shaped channel of AAR (Fig. 4a, Supplementary Fig. 10a). The CoA moiety occupies the same cleft in which NADPH was observed to bind in the AAR$_{NADPH}$–ADO structure (Supplementary Fig. 10b). The stearoyl-CoA and NADPH possess the

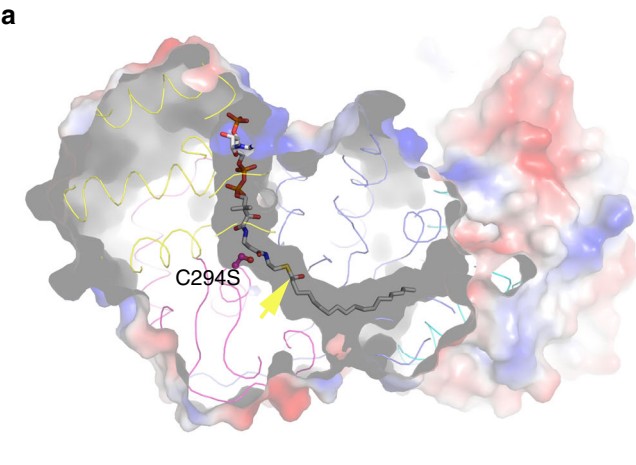

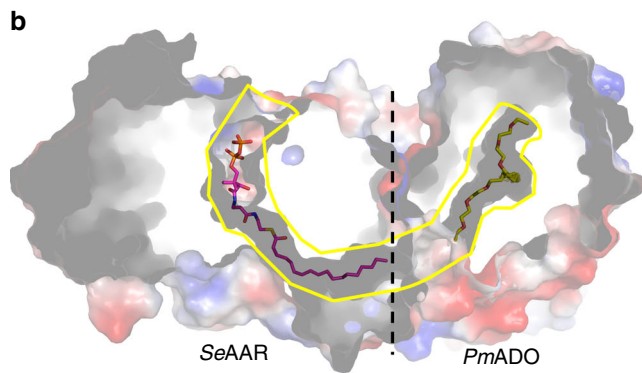

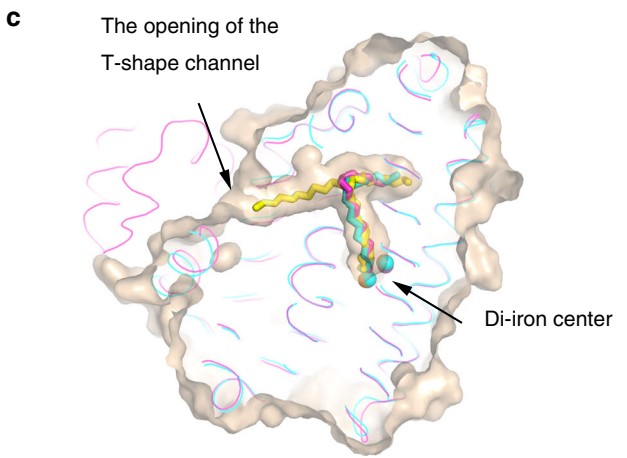

**Fig. 4 The AAR$_{stearoyl-CoA}$–ADO structure. a** Surface representation of AAR$_{stearoyl-CoA}$–ADO structure. Stearoyl-CoA is located in the L-shaped channel and shown as sticks. The residue C294S is not able to form thioester bond with the C1 atom (indicated by yellow arrow) of the stearic chain and is ~6 Å away from it. **b** Surface representation of SeAAR-PmADO model. In PmADO structure (PDB code 4PGI), two ligands (shown as sticks and colored yellow for the carbon atoms) are together located in the T-shaped channel, which has a big opening at the molecular surface. The tail end of stearic chain in SeAAR (magenta stick) points toward the entrance of the entering tunnel in PmADO. The black dashed line indicates the interface between AAR and ADO. The contiguous channel throughout AAR and ADO is highlighted by yellow line. **c** The superposition of PmADO structure (PDB code 4PGI) onto the ADO part of SeAAR-SeADO structure solved in this study and the free SeADO structure solved previously (PDB code 4RC5) to show the T-shaped channel. The PmADO structure (PDB code 4PGI) is shown in surface mode, and the ligand bound in the T-shaped channel is shown as yellow sticks. The SeAAR-SeADO structure (colored magenta) and the free SeADO structure (PDB code 4RC5; colored cyan) are shown in ribbon, with the bound ligand shown as sticks.

same ADP moiety, with the ribose phosphate group located at a slightly different position (Supplementary Fig. 10c). Our structures showed that their ADP parts are well superposed. The 3′-ribose phosphate group of CoA almost overlapped on the 2′-ribose phosphate of NADPH (Supplementary Fig. 10c), therefore is able to interact with the three positively charged residues of AAR. These results suggested that NADPH and CoA occupy the same site within AAR. In agreement with our structural observation, our binding assay showed that the wild-type AAR binds NADPH with a $K_D$ value of ~7.99 μM, whereas the C294S mutant is unable to bind NADPH after incubated with stearoyl-CoA (Supplementary Fig. 11), suggesting the unreleased stearoyl-CoA occupies the NADPH binding site. Our results are in line with previously suggested "ping-pong" catalytic mechanism employed by AAR[15,18].

It is noteworthy that stearoyl-CoA adopts an extended conformation, resulting in a shift of the aliphatic chain of ~6 Å

toward the AAR–ADO interface compared with that in other two AAR–ADO complex structures described above. The main cause of this shift is that the C294S mutant is not able to form a thioester bond, and thus fails to fix the stearic chain at the active site of AAR (Fig. 4a). This extended conformation of stearoyl-CoA induces movement of the Fr-I region of AAR (Supplementary Fig. 12a). Thus, Y23 and Y26, which form the terminal end of the acyl-tunnel in the AAR$_{thioester}$–ADO structure, are distant from the stearic moiety in the AAR$_{stearoyl-CoA}$–ADO structure. These changes result in a wider opening at the surface of the AAR molecule (Supplementary Fig. 12b). Furthermore, M21, Y23, and Y26 of AAR form additional hydrogen bonds with R218 and M221 from helix 8 of ADO (Supplementary Fig. 12c), and the conformation of M221 of ADO alters so that it no longer seals the acyl-tunnel of AAR as it does in the AAR$_{thioester}$–ADO structure (Supplementary Fig. 12b). Together, these structural changes facilitate the release of produced aldehydes from AAR, suggesting that the AAR$_{stearoyl-CoA}$–ADO structure mimics the aldehyde-releasing state of AAR, in which the free aldehyde is released, ready to be transferred to ADO.

**A potential aldehyde-transferring channel from AAR to ADO.** Previous studies[18], together with our enzymatic assay, demonstrated that both AAR and ADO alone are functional (Table 3). In addition, we found that when both incubated with AAR, the wild-type ADO yields two times more product (heptadecane) than the triple mutant (E196A/E200A/D201A) of ADO (Supplementary Fig. 13), which is unable to form the AAR–ADO complex (Table 2, Supplementary Fig. 5). These results together indicate that the formation of AAR–ADO complex is to facilitate the direct transfer of the aldehyde product from AAR to ADO, which increases the efficiency of alkane production in this pathway. Direct transfer would require a contiguous channel from AAR to ADO. In our AAR$_{ligand}$–ADO structures, the acyl-tunnel of AAR has an opening at the surface of AAR and toward AAR–ADO interface (Supplementary Fig. 12b), suggesting the release direction of the aldehyde. However, in these structures, ADO exhibits an occluded conformation, therefore, we were not able to identify the entry point for the aldehyde to access the active site of ADO based on our AAR–ADO complex structures. Previously reported crystal structures of PmADO showed that a T-shaped channel is present inside ADO, with an opening between helices 7 and 8 at the ADO surface, which was suggested to represent the entry point for aldehyde[20]. The helices 7 and 8 of

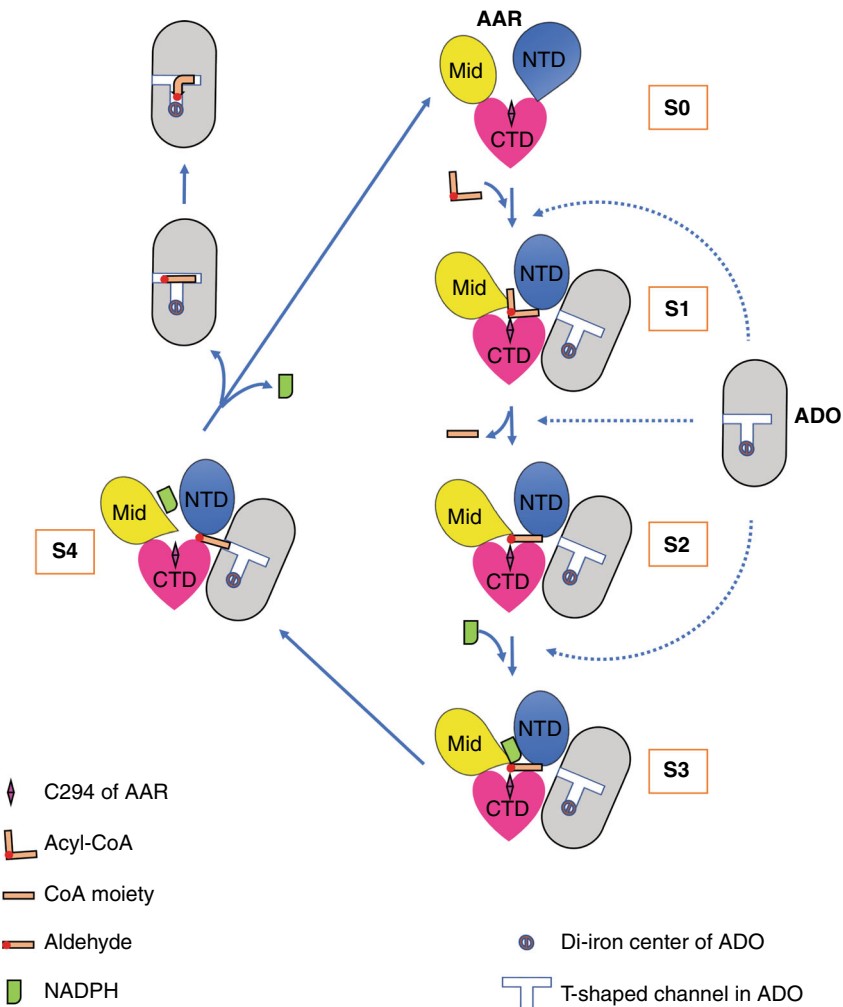

**Fig. 5 Cartoon model of the proposed catalytic process of AAR in the two-step pathway.** The NTD, mid-domain, and CTD of AAR are colored blue, yellow, and magenta, respectively. ADO is colored gray. Note that ADO may associate with AAR at several different states (such as S1, S2, and S3), since its binding is for assisting the aldehyde transferring, but not essential for substrate/NADPH binding.

ADO are at the AAR–ADO interface and helix 7 is essential for the complex formation. On the basis of these results, we superposed the *Pm*ADO structure (PDB code: 4PGI) with the ADO part of our AAR$_{stearoyl-CoA}$–ADO structure, and found that the two openings of both *Se*AAR and *Pm*ADO face each other (Fig. 4b), thus the acyl-tunnel of AAR and the T-shaped channel inside ADO are linked, forming a long contiguous channel in the superposed model. The opening at ADO surface leads to the entering tunnel, which corresponds to one half of the horizontal arm of the T-shaped channel. The other half of the horizontal arm together with the vertical arm forms the substrate cavity, which was observed in several ADO structures solved in both the present study, as well as in earlier reports[19,21,22] (Fig. 4c). Based on these structural observations, here we propose a model for the process of transferring aldehyde from AAR to ADO. The alkyl group enters the T-shaped channel of ADO, as a result of which the entire aldehyde molecule passes through the entering tunnel, before it reaches the distal end of the whole horizontal arm of the T-shaped channel. The aldehyde group proceeds toward the inside the horizontal arm after the transfer is completed, and upon the bending of the alkyl chain in the middle, accesses the di-iron center of ADO (Fig. 5).

In our structures, W178 of *Se*ADO covers the entrance of the ADO substrate entering tunnel, while R191 represents the corresponding residue in *Pm*ADO. In *Pm*ADO structure,

R191 switches its rotamer and as a result, opens the entry of the entering tunnel (PDB code: 4PGI; Supplementary Fig. 14a). When we superposed the structure of *Pm*ADO (PDB code: 4PGI) on *Se*ADO in our complex structure, we found that W178 in *Se*ADO and R191 in *Pm*ADO are located at the same positions, with their main chain atoms and $C_\beta$ atoms well superposed (Supplementary Fig. 14b). Our structural comparison indicated that W178 could easily change its rotamer to swing its indole ring away, thus opening the entry without inducing large movement of the main chain atoms. Together, these results suggest that W178 (or R191 of *Pm*ADO) functions as a gate residue; it may shield the hydrophobic entering tunnel of ADO from the aqueous environment, whereas the approaching aldehyde produced by AAR may result in its side chain swinging away, leading to the opening of the entry at the ADO surface. These conformational changes result in the formation of a long hydrophobic channel through which the aldehyde could be transferred from AAR to ADO.

Our structures provide strong evidence that the interactions between the NTD of AAR and the C-terminal region of ADO are essential for the complex formation and the aldehyde transfer. This is consistent with the previous analysis showing that introducing the ADO–AAR fusion protein (C-terminus of ADO linked with N-terminus of AAR) into *Escherichia coli* increased its alkane yield by 4.8-fold compared with the strain expressing both AAR and ADO, but separately[8]. The fused protein may

easily form a complex similar to that presented here, and promote alkane production by facilitating the direct transfer of the aldehyde product from AAR to ADO.

**Structural variations of AAR in active and resting states.** Superposition of the $AAR_{apo}$ structure onto the AAR part of the AAR–ADO complex structures showed that the Fr-II region, which is flexible in the $AAR_{apo}$ structure (Supplementary Fig. 3a), is stabilized by the hydrogen bond between R118 in Fr-II and D201 of ADO (Fig. 1c). In addition, three other regions of AAR, namely Fr-I, Fr-III, and Fr-IV (Q277–M284), exhibit dramatic conformational changes between the structures of apo- and ligand-bound states (Supplementary Fig. 15a). The N-terminal segment (H7–A37) containing Fr-I is more loosely packed, potentially hindering AAR–ADO complex formation in the $AAR_{apo}$ structure. Nevertheless, it adopts a conformation that facilitates intermolecular interactions in the AAR–ADO complex structures (Supplementary Fig. 15a). The Fr-III region (W221–T235) in the mid-domain, which is involved in NADPH binding, swings toward the central cleft in the AAR–ADO complex structures, resulting in a better fit of NADPH in the cleft (Supplementary Fig. 15b). Moreover, a short fragment containing M279 in the CTD (Fr-IV), which potentially clashes with the stearic chain in the $AAR_{apo}$ structure (Supplementary Fig. 15c), changes its conformation to shape the acyl-tunnel in the AAR–ADO complex structures (Supplementary Figs. 7c and 15c). The three fragments tend to move toward the active center, creating a more compact architecture of AAR within the AAR–ADO complex structures (Supplementary Fig. 15a). The compact folding allows AAR to better accommodate the substrate/NADPH, therefore AAR in the AAR–ADO complex structures is likely to adopt an active conformation, which allows the catalytic reaction to proceed, whereas the $AAR_{apo}$ structure represents the resting state. Furthermore, our structures suggest that the active form (the ligand-bound form) is more favorable for AAR to bind ADO, compared with the loosely packed apo form of AAR.

In addition, our structural analysis suggested a pivotal role of Y247 for maintaining AAR in a compact conformation. It forms a hydrogen bond with H7 from the NTD and hydrophobic interactions, with aromatic residues from both CTD and NTD of AAR (Supplementary Fig. 15d), thus drawing the mid-domain, CTD, and NTD together. In addition, Y247 may also be involved in the catalytic process, as it participates in the formation of the acyl-tunnel via its benzene ring and stabilizes NADPH via its carbonyl group (Fig. 3b). In agreement with these structural observations, the Y247F mutant of AAR exhibited reduced binding affinity with stearoyl-CoA, and maintained only ~50% of the catalytic activity of the wild type (Table 3). Furthermore, Y247A completely failed to bind stearoyl-CoA (Table 3, Supplementary Fig. 9). Interestingly, we found that Y247A mutant also loses the capability to bind ADO, while the binding affinity of Y247F mutant with ADO is almost the same as that of AAR wild type (Table 2, Supplementary Fig. 6). This observation suggests that mutation of the Tyr residue to Ala induces the structural or conformational change of the whole AAR protein, resulting in the complete loss of its function.

## Discussion

We solved the crystal structures of AAR and the AAR–ADO complex bound with several, structurally different ligands, thus greatly advancing the understanding of the detailed mechanisms of the catalytic cycle of AAR(–ADO) responsible for alkane production. Interestingly, we noticed that neither the AAR protein with ligands nor the AAR–ADO complex without ligands yielded

crystals, implying that both types of binary complexes are flexible, and only the ternary complex (AAR–ligand–ADO) is sufficiently stable to yield crystals. These results suggest that AAR shows a synergistic effect of substrate binding and ADO association. In agreement with this observation, our structural data suggested that the compact folding of AAR in the complex is favorable for substrate/NADPH binding and ADO association, and suggest that the formation of AAR–ADO complex facilitates the catalysis of AAR by relieving its substrate/product inhibition[18].

However, the stable AAR–ADO complex may not represent the optimal state for alkane production, since the rate-limiting enzyme ADO with low turnover may slow down the whole catalytic process of this pathway, if it is stably complexed with AAR. In agreement with this idea, it was previously shown that alkane yield is upregulated in *E. coli* when expressing ADO and AAR at an increased molar ratio of 3:1 compared with the control strain (1:1 molar ratio of ADO and AAR)[8,27]. It was earlier reported that both AAR and ADO alone exhibit full activity, thus complex formation is primarily responsible for the subsequent aldehyde transfer[18]. We assumed that after the transfer process is completed, AAR returns to the apo state and adopts a loose conformation similar to the $AAR_{apo}$ structure, and dissociates from ADO. As a result, the free AAR could reload its substrate and bind with another ADO molecule, thus the rate-limiting effect of ADO could be partially relieved. The transient AAR–ADO complex may be critical for optimizing the catalytic efficiency of this pathway.

Basing on these assumptions, together with the insights from the AAR structures captured at different states, we propose a five-state (S0–S4) model that could account for the catalytic mechanism of AAR (Fig. 5). AAR alone adopts a loose conformation as shown in the $AAR_{apo}$ structure, which represents the resting state (ligand-free) of AAR (S0). The first step of catalysis involves the binding of either acyl-ACP or acyl-CoA substrate (S1). This process is stimulated by monovalent ion atoms, such as $K^+$ ions. Simultaneously, an acyl-enzyme intermediate is spontaneously formed before either the ACP or CoA moiety is released (S2, represented by the $AAR_{thioester}$–ADO structure). Our results indicated that substrate binding induces a conformational change of AAR, resulting in its compact folding, facilitating the formation of a stable $AAR_{ligand}$–ADO complex in S1 and S2 states. In the complex, the acyl moiety is linked with C294 at the center of AAR molecule, whereas the alkyl group is pointing toward the AAR–ADO interface. Subsequently, NADPH is incorporated into the cleft above C294 of AAR (S3, represented by the $AAR_{NADPH}$–ADO structure), before AAR catalyzes the hydride transfer of NADPH, and before reduction of the acyl moiety to a long-chain aldehyde. It is noteworthy that AAR in apo state (S0) is able to associate with ADO, as shown by the ITC results in the present study as well as in earlier report[18]. The AAR–ADO binary complex may further bind the substrate, resulting in a stable ternary complex. Moreover, AAR alone is fully active and capable of catalyzing the conversion of acyl-ACP/acyl-CoA to aldehyde, and the formation of AAR–ADO complex is to fulfill the aldehyde transfer function. As a result, AAR may form a complex with ADO at several different states, even before binding substrate (S0 state) or after binding NADPH (S3 state). Following aldehyde production and complex formation, the free aldehyde product is released from AAR (S4, represented by the $AAR_{stearoyl-CoA}$–ADO structure), which results in further conformational changes of the N-terminal region of AAR at the AAR–ADO interface. This conformational change of AAR and the moving aldehyde may lead to the altered conformation of the C-terminal residues of ADO, such as the gate residue (W178 in *Se*ADO and R191 in *Pm*ADO), thus opening the entering tunnel in ADO. After aldehyde transfer into ADO, AAR may return to

the resting state exhibiting a loose conformation (S0), detach from ADO, and prepare for the next cycle. The aldehyde in ADO moves further to occupy the substrate cavity with its head group close to the di-iron center of ADO (Fig. 5). Our suggested mechanism explains the increased alkane yield when providing ADO, especially in higher molar ratio to AAR (ref. [27]), and highlights the functional importance of the reversible association and dissociation of AAR and ADO in the alkane biosynthesis pathway.

## Methods

**Protein expression and purification.** The AAR gene (Synpcc7942_1594, gene ID:3775018) from the complete genome of Synechococcus elongatus PCC 7942 (Genbank ID:CP000100.1), was synthesized (GeneScript, China) and cloned (Supplementary Tables 1 and 2) into the modified version of vector pET24a (Novagen), that encodes a tobacco etch virus (TEV) protease cleavage site with the sequence of ENLYFQG, followed by a downstream 21-residue peptide, including a His(6)-tag with the sequence of SMNSSSVDKLAAALEHHHHHH. Upon verification of the correct sequence, the plasmid containing AAR gene was transformed into E. coli strain BL21(DE3) (TransGen Biotech) along with the plasmid pG-KJE8 (TaKaRa), which encodes five molecular chaperones of dnaK, dnaJ, grpE, groES, and groEL. The transformed cells were cultured in Luria–Bertani medium containing 50 μg mL$^{-1}$ kanamycin, 34 μg mL$^{-1}$ chloramphenicol, 2 mg mL$^{-1}$ L-arabinose, and 5 ng mL$^{-1}$ tetracycline in shake flasks at 37 °C.When the optical absorption density at 600 nm (OD$_{600}$) reached 0.8–1, the expression of AAR was induced by addition of 1 mM IPTG (isopropyl β-D-1-thiogalactopyranoside) and continued shaking at ~200 rpm for overnight at 20 °C. The cells were harvested and disrupted by ultra-sonication in lysis buffer containing 50 mM Tris-HCl, pH8.0, 500 mM NaCl, 1 mM TCEP (tris (2-carboxyethyl)phosphine), 10% glycerol, 1 mM PMSF (phenylmethylsulfonyl fluoride), 5 mM KCl, and 2 mM MgCl$_2$ and 10 mM imidazole. Cell debris was discarded after centrifugation at 38,000 × g for 30 min at 4 °C. The supernatant containing the expressed AAR protein with His-tag (AAR-His) was loaded onto a Ni affinity column (GE Healthcare) pre-equilibrated with lysis buffer. The AAR-His protein was then washed and eluted with the lysis buffer containing 50 mM and 300 mM imidazole, respectively. To obtain the His-tag-free AAR protein, AAR-His eluted from the Ni affinity column was treated with TEV protease for 12 h at 4 °C, and then flown through the Ni affinity column. This treatment removes the C-terminal 22-residue peptide, including the His-tag, encoded by the vector, thus results in the AAR protein alone (AAR). Both AAR-His and AAR proteins were further purified through size-exclusion chromatography with elution buffer of 10 mM HEPES (pH 7.5), 50 mM NaCl, 5 mM KCl, 2 mM MgSO$_4$, and 1 mM TCEP. The protein was concentrated to ~30 mg mL$^{-1}$ through measuring its absorption at 280 nm using a calculated extinction coefficient of 43,430 M$^{-1}$ cm$^{-1}$ (http://ca.expasy.org). SeADO gene (Synpcc7942_1593, gene ID: 3775017) from the complete genome of Synechococcus elongatus PCC 7942 (Genbank ID:CP000100.1) was constructed (Supplementary Tables 1 and 2) and expressed similarly as that of SeAAR gene, except that the chaperone-encoded plasmid pG-KJE8 (TaKaRa) was not used. The expression of ADO protein was induced by adding 1 mM IPTG, 1 mM FeSO$_4$, and shaken at 30 °C for 5 h. The cells were disrupted by ultra-sonication in lysis buffer containing 20 mM Tris-HCl, pH8.0, 1 mM TCEP, 500 mM NaCl, and 10 mM imidazole. The supernatant after centrifugation was loaded onto a Ni affinity column (GE Healthcare) pre-equilibrated with lysis buffer. The ADO protein was washed and eluted with the lysis buffer containing 50 mM and 300 mM imidazole, respectively. The eluted ADO protein was treated with TEV protease overnight at 4 °C, and then flown through the Ni affinity column to remove the His-tag, and further purified through size-exclusion chromatography with elution buffer containing 20 mM Tris-HCl, pH8.0, and 1 mM TCEP. All mutant plasmids used in this experiment were constructed with the pET24a plasmid of wide type, namely pET24a-AAR and pET24a-ADO by PCR (polymerase chain reaction), with primer sequences shown in Supplementary Table 2. The mutations of the target gene were verified by sequencing. The over-expressed AAR and ADO mutant proteins were purified using the same protocol as the wide-type AAR and ADO proteins, respectively. The purity of all the wild type and mutant forms of AAR and ADO are verified through SDS–PAGE.

**ITC measurement.** The binding affinity of AAR with ADO or stearoyl-CoA was measured by isothermal titration calorimetry (ITC) using a MicroCal Auto-ITC200 microcalorimeter (GE Healthcare Bio-sciences, Pittsburgh, PA). To measure the affinity between AAR and ADO, all the protein samples were prepared in the same buffer containing 25 mM Tris-HCl, pH 7.5, and 1 mM TCEP. Wild type or mutants of ADO (or AAR) were placed into the sample cell (200 μL working volume) at a final concentration of 30 μM (60 μM for AAR) and was titrated with successive 2-μL injections of AAR protein with concentration of 500 μM (or ADO protein with concentration of 1 mM). Titrating AAR (500 μM) or ADO (1 mM) into the same buffer was performed as a control to determine the background correction. For measuring the affinity of AAR or its mutants with stearoyl-CoA, and evaluating the effect of monovalent ion on their binding, 30 μM AAR was titrated against stearoyl-CoA (500 μM for the presence of KCl and 650 μM for the absence of KCl) using 3-μL injections in buffer A (25 mM Tris-HCl, pH 7.6 and 1 mM TCEP) in

the presence or absence of 200 mM KCl. Injection of stearoyl-CoA into the reaction buffer was performed as control to determine background correction. The data were normalized and processed with the Origin7 software package (GE Healthcare Bio-sciences, Pittsburgh, PA).

**Microscale thermophoresis measurement.** The binding affinity of AAR with NADPH was measured by microscale thermophoresis (MST) experiment, which was performed on a Monolith NT.labelfree system (NanoTemper Technologies, Germany). Proteins of wild type and C294S mutant of AAR were diluted at a stock concentration of 1 μM and 4 μM, respectively, and centrifuged at 15,000 rpm for 10 min. The dilution buffer contains 20 mM Tris-HCl pH 7.5, 0.2 M KCl, 1 mM TCEP, and 0.5 mM MgCl$_2$. A two-fold dilution series was prepared for ligand NADPH with dilution buffer. The experiment about ligand NADPH was accompanied by a step-by-step gradient dilution process. Subsequently, 10 μL of AAR protein (wild type or C294S mutant) was mixed with 10 μL ligands with different concentration. Samples were filled into hydrophilic capillaries (Monolith NT. labelfree capillary). The label free MST assay was performed with 30% excitation power and the high MST power controlled by the MO.Control Software Version 1.6.1. $K_D$ fit function of Mo.Affinity Analysis v2.3 was used to fit curve and calculate the value of dissociation constant ($K_D$).

**Enzymatic assay.** For measuring the enzymatic activity of AAR, the AAR protein at the concentration of 20 μM was mixed with 2 mM NADPH and 0.5 mM stearoyl-CoA in a 100 μL reaction buffer containing 20 mM HEPES pH 7.5, 200 mM KCl, 0.5 mM MgCl$_2$, and 0.5 mM TCEP. For measuring the enzymatic activity of AAR–ADO complex, additional 40 μM ADO, 150 μM ferrodoxin, and 50 μM Fd-NADP$^+$ oxidoreductase (FNR) were added to the AAR reaction buffer, with increased NADPH concentration of 4 mM. The ferrodoxin and FNR are recombinant proteins from cyanobacterial strains Synechocystis sp. PCC 6803 and Synechocystis sp. PCC 7942, respectively, and were added into the reaction buffer for supplying electrons for the reduction of ADO. The mixture was incubated at 30 °C for 5 h. The produced octadecanal and heptadecane were extracted by addition of an equal volume of ethyl acetate (Sigma Aldrich, HPLC grade) and centrifuged at 13,000 × g for 15 min at room temperature. The extraction step was repeated for three times. The extracted supernatant was pooled together, dried with nitrogen gas, and then resolved in 20 μL ethyl acetate for the GC-MS measurement.

The concentration of octadecanal and heptadecane was measured using an Agilent 7890 A gas chromatograph coupled with an Agilent 7000B QQQ mass spectrometer. A total of 2 μL sample was loaded with the helium as the carrier gas onto the chromatographic column HP-5ms (30 mM × 0.25 mM ID, 0.25 μm; Agilent) at the flow rate of 1.3 ml min$^{-1}$. The injector temperature maintained at 300 °C with spotless mode. The GC oven settings were as follows: initial temperature 75 °C, held for 5 min, then at 40 °C min$^{-1}$, increased to 325 °C and held for 5 min. Ionization mode of EI (70 eV, 230 °C) and full scan of 50–550 m/z were chosen for MS detection. The standard curve was plotted based on the measuring of the commercial octadecanal and heptadecane (Sigma Aldrich) at a gradient concentration (from 3.125 to 75 μg mL$^{-1}$). Data were acquired and processed using MassHunter Workstation Software and the NIST database.

**Preparation of the AAR–ADO complex.** To get the AAR in complex with stearoyl-CoA and ADO, AAR was first incubated with its substrate stearoyl-CoA at the molar ratio of 1:3 for 30 min at 18 °C, and was further incubated with ADO protein at 1:1 molar ratio for additional 4 h. The unbound stearoyl-CoA was removed from the AAR–ADO complex through gel filtration using the superdex200 10/300GL column, with elution buffer containing 10 mM HEPES (pH 7.2), 50 mM NaCl, 5 mM KCl, 2 mM MgSO$_4$, and 1 mM TCEP. The eluted protein complex was characterized through SDS–PAGE and concentrated to 40 mg mL$^{-1}$ for crystallization. The molecular weight of both the AAR protein alone and AAR protein covalently bound with stearoyl-CoA were determined by mass spectrometry (HPLC-Q-TOF-MS).

**Crystallization, structure determination, and refinement.** Crystallization screens were carried out at 18 °C by mixing equal volume of protein and reservoir solution, using the sitting-drop vapor diffusion method. Crystals of AAR-His were grown in condition of 0.1 M Tris-HCl, pH8.5, and 1.2 M NaAc. For phasing, the AAR-His crystals were soaked in mother liquor (0.1 M Tris-HCl, pH8.5, 200 mM NaAc, and 25% PEG3350) containing 10 mM ethyl mercury phosphate for 10 s. Crystals of AAR were harvested in reservoir solution of 0.1 M citric acid, pH6.0, and 1 M (NH$_4$)$_2$SO$_4$. The crystals of AAR–ADO in complex with substrate (AAR$_{thioester}$–ADO and AAR$_{stearoyl-CoA}$–ADO) were harvested in reservoir solution containing 0.1 M HEPES, pH 7.5, and 20% PEG8000. To obtain the crystal of the AAR$_{NADPH}$–ADO, the crystals of AAR$_{thioester}$–ADO were soaked with 1 mM NADPH in reservoir solution overnight. Crystals were flash-cooled in liquid nitrogen at 100 K for data collection using 2.5 M NaAc in mother liquor as cryo-protectant for AAR-His crystals, 20% glycerol in mother liquor as cryo-protectant for AAR$_{apo}$ crystals, and the reservoir solution as cryo-protectant for AAR–ADO complex crystals. Diffraction data were collected at Shanghai Synchrotron Radiation Facility at wavelength of 0.98 Å (0.9779~0.9792) Å (ref. [28]). Diffraction data were processed and scaled with HKL-2000 package[29]. The initial phase of AAR-His

was obtained from the Hg-derivative crystals by single isomorphous replacement anomalous scattering method using AutoSol in PHENIX package[30]. In the AAR-His structure, the C-terminal peptide encoded by the vector forms an α-helix, which is participating the molecular packing in the crystals, leading to an artifact of AAR structure. Thus, the AAR-His structure was used only for phasing and all other experiments were performed using AAR (without C-terminal 20-residue-peptide) protein. The structure of $AAR_{apo}$ was solved by Molecular Replacement using the structure of AAR-His as a search model. The $AAR_{ligands}$–ADO structures were solved by Molecular Replacement using the structures of AAR-His and ADO (PDB code:4RC8) as search models. Most part of the structures were built by AutoBuild in PHENIX package[30], whereas a small parts were built manually by Coot[31]. The models were refined by Phenix.refine[30]. A summary of data collection and structure refinement statistics is provided in Supplementary Table 1. The Ramachandran plot showed that most of the residues (95–96.4%) in these four structures are in the most favored regions, and no residues lie in the disallowed region. All the figures of the structures were prepared by Pymol (The PyMOL Molecular Graphics System, Version 1.3r1, Schrodinger, LLC., 2010). The difference map for showing the density of ligands were calculated by the program "create a map from map coefficients" in phenix supplying the PDB without ligands and MTZ files.

**Reporting summary**. Further information on experimental design is available in the Nature Research Reporting Summary linked to this paper.

## Data availability

The refined models of AAR and AAR–ADO complexes have been deposited in the Protein Data Bank (www.rcsb.org) with PDB codes 6JZQ, 6JZU, 6JZY, and 6JZZ for $AAR_{apo}$, $AAR_{thioester}$–ADO, $AAR_{NADPH}$–ADO, and $AAR_{Stearoyl-CoA}$–ADO, respectively. The source data underlying Table 3 and Supplementary Figs. 4, 11, and 13 are provided as a Source Data file. Other data supporting the findings of this study are available from the corresponding author upon reasonable request.

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

## Acknowledgements

We are grateful to the staffs at the Shanghai Synchrotron Radiation Facility (Shanghai, China) for technical support during diffraction data collection. We thank Y. Chen, Z. Yang, and B. Zhou from IBP, CAS for ITC experiment; N. Zhu from IBP, CAS for GC/MS experiment; L. Niu, X. Ding, M. Zhang, and F. Yang from IBP, CAS for mass spectrometry; X. Lu, W. Wang, and Y. Lin from Qingdao Institute of Bioenergy and Bioprocess Technology, CAS for providing the SeADO plasmid and valuable discussions; L. Wang from Institute of Botany, CAS and Y. Wu from the Institute of Microbiology, CAS for MST experiment. The authors thank Dr. Torsten Juelich (Peking University, China) for linguistic assistance during the preparation of this manuscript. The project was funded by the Strategic Priority Research Program of CAS (XDB27020106 and XDB08020302), the National Key R&D Program of China (2017YFA0503702 and 2011CBA00900), National Natural Science Foundation of China (31770778 and 31930064), and the Key Research Program of Frontier Sciences of CAS (QYZDB-SSW-SMC005).

## Author contributions

M.L. and W.C. conceived the project; Y.G. and C.J. did the plasmid construction; Y.G. expressed, purified AAR and ADO, prepared AAR–ADO complex and did the crystallization; L.S. and X.Z. assisted the protein expression and purification; H.Z., Y.G., P.C., and X.P. collected diffraction data; M.F. determined the AAR-His structure; H.Z solved, built, and refined the $AAR_{apo}$ and AAR–ADO complex structures; M.L. and H.Z. analyzed the structures, and wrote the manuscript; all authors discussed and commented on the results and the manuscript.

## Competing interests

The authors declare no competing interests.
