## [Peer Review File · Nature Communications]

Reviewers' comments:

Reviewer #1 (Remarks to the Author):

The manuscript by Gao et al. examines the catalytic mechanism of cyanobacterial acyl-acyl carrier protein reductase, AAR, using AAR and ADO enzyme complexed structures. The protein structural analyses for the complexed crystals are to the best of my knowledge technically sound and performed correctly. However, several questions are raised in the interpretation of the data results.

Major concerns:

The authors suggest the catalytic process of AAR through the protein structural analyses and biochemical data. In this proposed mechanism, although active machinery has been explained through the analysis of protein structure and biochemical data, it does not appear to have clear information on when and how ADO separation and binding with AAR are performed. When the substrate is transferred from AAR to ADO, it may be necessary to elaborate the position change of the substrate, described as "rotameric switch" in this paper.

Has the author tested the enzymatic activity of the AAR-ADO complexed crystals?

What is the gene ID for AAR and ADO?

Correction:

In fig S9, change NpADO to PmADO

Reviewer #2 (Remarks to the Author):

Biosynthesis of alkanes in cyanobacteria has attracted great attention as a substitute for petroleum-based fuels. Cyanobacteria produces alkanes using a two-step reaction involving two enzymes, AAR and ADO. Although the crystal structures of ADO have been already known, structure of AAR has been elusive. This paper for the first time reports the structure of AAR and those of the AAR-ADO complexes in which AARs bind various ligands. The results suggest the structures of the AAR-cofactor and AAR-substrate complexes and a possible channel from AAR to ADO for efficient delivery of an aldehyde. Finally, the authors proposed a catalytic cycle involving structural changes of AAR and binding/dissociation of ADO. Although the purpose of this work is important and interesting, there are many concerns as described below.

Major concerns:

1. One of the major concerns on this paper is the direction of the acyl chain in the acyl tunnel of AAR. The present crystal structures showed that the alkyl group, instead of the aldehyde group, of the acyl chain is pointed toward the inside of ADO. If so, however, the alkyl group enters deep inside the ADO protein, where the reactive center is located, and the aldehyde group of the acyl chain cannot reach the reactive center of ADO. This is unreasonable. Thus, the present results cannot clearly explain the mechanism of efficient delivery of an aldehyde product from AAR to ADO.

2. Although AAR was previously suggested to have higher activity for acyl-CoA than for acyl-ACP (ref. 15), it was later demonstrated that AAR has a strong preference for acyl-ACP substrates over acyl-CoA substrates. Since the natural substrate of AAR is acyl-ACP, the authors should use acyl-ACP rather than acyl-CoA for crystallization studies. Without the crystal structure of the complex between AAR and acyl-ACP, the mechanism of AAR catalysis cannot be solved. In this sense, the present report is preliminary and needs additional data.

3. Unfortunately, the binding site of Mg²⁺ ion(s) was not determined in the present study,

although this information is important in understanding the role of Mg²⁺ in AAR catalysis.

4. The authors assumed that Mg²⁺ and K⁺ ions bind to AAR in the crystals. However, there have been no biochemical evidence showing that the metal ions do bind AAR. Therefore, it is required to confirm that Mg²⁺ and/or K⁺ really bind to AAR using ITC or other methods. Note that it is possible that K⁺ ions do not directly bind AAR but just enhance the solubility of acyl-CoA in the presence of Mg²⁺ ions. See Constantinides & Steim (1986) Arch Biochem Biophys. 250(1): 267-270.

5. It is also unfortunate that a long contiguous channel from AAR to ADO was not observed in the present crystal structures.

6. To understand the activity of AAR and its mutants, it is necessary to measure *k*_{cat} and *K*_m values, instead of reporting just an octadecanal yield (see Table S2). Also, it is desirable to measure activities of all mutants used in the present study.

7. To verify the AAR-ADO interface, the authors created three ADO mutants targeting the acidic residues of helix 7 and measured the binding affinity of AAR with them using ITC. The results showed that both the E200A/D201A double mutant and the E196A/E200A/D201A triple mutant of ADO completely lost the binding affinity to AAR. However, it is still unknown which of E200 and D201 is essential for the AAR binding. In addition, contributions of E196 and E208 are unknown. To clarify these issues, it is necessary to create four additional mutants, namely the E196A, E200, D201A and E208A single mutants and study the contribution of each acidic residue of ADO on the AAR binding using ITC. It is also recommended to make some AAR mutants at positively charged residues, to confirm the importance of electrostatic interactions between AAR and ADO.

8. The present results indicate that acyl-CoA and NADPH binds at the same site on AAR competitively. To verify this, it is required to confirm that acyl-CoA cannot bind to AAR simultaneously with NADPH using, for example, ITC.

9. It is required to measure the binding affinity (*K*_d) of ADO with the Y247A and Y247F mutants of AAR, because Y247 of AAR forms a hydrogen bond with H7 of ADO.

Other comments:

10. In Fig. S9, W178 in SeADO is compared with R191 in PmADO. For this purpose, the authors used the PDB file of 4PGK for the structure of PmADO. However, this structure has the R191L mutation (not Val but Leu) and is not appropriate for the above purpose. The authors should use other PmADO structures that do not have mutation at R191 in Fig. S9 and Fig. 4.

11. To solve the AAR(NADPH)-ADO structure, the authors soaked the AAR(thioester)-ADO crystal with NADPH. However, it is curious why the authors did not try to construct the AAR-ADO-NADPH ternary complex without addition of acyl-CoA.

12. When presenting the ITC data, the authors only showed the *K*_d values (see Table S2 and Figs. S3 and S6). However, to check the validity of the ITC experiments, other parameters should also be shown, including ΔH , ΔS , and stoichiometry.

13. The ITC data on the binding of AAR with stearyl-CoA, shown in Fig. S6, indicate a stoichiometry of 1:0.5, but this value is not reasonable.

14. In the ITC measurements, the authors prepared an AAR sample of 720 μ M. However, since AAR is prone to aggregation (see ref. 15), there is a concern that AAR was aggregated, and the ITC measurements may not be properly carried out. If the authors have some tips to solubilize the AAR protein at high concentrations, please describe them in detail.

15. The methods section should be described in more detail in Supplementary Information. To reproduce the purification of the AAR and ADO samples, it is required to describe, for example, components of media and antibiotics, timing of IPTG induction, cultivation time of *E. coli*, methods of determination of protein concentration and so on. In addition, please describe the amino acid sequence (22 residues) attached at the C-terminal region of AAR as a His-tag.

16. The purity of the AAR and ADO proteins should be shown. For example, the data on SDS-PAGE after purification and the profiles of size exclusion chromatography using Superdex 200 should be attached in Supplementary Information.

17. In the AAR(apo) structure, there were three flexible regions, called Fr-I, Fr-II and Fr-III. Although roles of Fr-I and Fr-III are discussed in the manuscript, nothing is discussed on Fr-II.

18. Page 2, line 23-24: In the Abstract, the authors wrote that the two enzymes act in series and potentially form a complex that efficiently converts long chain fatty acids into hydrocarbons. However, this sentence is incorrect: instead of fatty acids, fatty acyl-ACP or fatty-acyl-CoA is converted into hydrocarbons.

19. The authors should clearly describe which cyanobacterial species AAR and ADO are derived from. Line 78 describes that AAR for the apo form is from *Synechococcus elongatus* PCC 7942. Is the same AAR used for AAR-ADO complexes? Is ADO derived from the same cyanobacterial species?

20. In Fig. S4c, the lines pointing to M21 are not properly shown.

21. There are many typographical errors in the manuscript:

- Line 78: *Synechococcus elongates* 7942  *Synechococcus elongatus* PCC 7942
- Line 333: plasmid GKJE8  plasmid pGKJE8
- In Fig. 2c, V8 is probably L8.
- Line 2 of Fig. S1 caption: homologous  homologues
- Line 2 of Fig. S1 caption: *elongatus* 7942  *elongatus* PCC 7942
- Line 3 of Fig. S1 caption: *Nostocacea* sp. PCC 7120  *Anabaena* sp. PCC 7120
- The title of Fig. S9: NpADO  PmADO
- Table S2: Kd value (uM)  Kd value (microM)

Response to Reviewer 1

The manuscript by Gao et al. examines the catalytic mechanism of cyanobacterial acyl-acyl carrier protein reductase, AAR, using AAR and ADO enzyme complexed structures. The protein structural analyses for the complexed crystals are to the best of my knowledge technically sound and performed correctly. However, several questions are raised in the interpretation of the data results.

We thank the reviewer for the time and efforts in reading and assessing our manuscript, and for the positive comments and thoughtful suggestions.

Major concerns:

1. The authors suggest the catalytic process of AAR through the protein structural analyses and biochemical data. In this proposed mechanism, although active machinery has been explained through the analysis of protein structure and biochemical data, it do not appear to have clear information on when and how ADO separation and binding with AAR are performed. When the substrate is transferred from AAR to ADO, it may be necessary to elaborate the position change of the substrate, described as “rotameric switch” in this paper.

Thanks for your suggestion. Our results indicated that apo-AAR is able to bind either ADO or stearyl-CoA, with K_d values of approximately 2.0~2.2 μM and 0.09 μM , respectively, indicating that both AAR-ADO and AAR-ligand binary complexes can be formed in solution. However, the binary complexes did not yield any crystals despite our extensive screening in the crystallization conditions. Only the ternary complex (AAR-ligand-ADO) is able to form crystals. These results suggest that the ternary complex is more stable than the binary complexes. Indeed, our structures showed that the conformation of apo-AAR is less favorable for interacting with ADO, whereas ligand-bound AAR (AAR_{ligand}) adopts a compact conformation, which is preferable for ADO binding. Based on these results, we suggested that the association and separation of ADO with AAR should be related with the substrate binding and product releasing from AAR, respectively. We proposed that ADO detaching from AAR is occurred after the completion of aldehyde transferring from AAR to ADO. The switching from ligand-bound state to apo state of AAR results in the relatively loose conformation of its NTD, which is responsible for interacting with ADO, and thus leads to the dissociation of ADO from AAR.

However, it is difficult to assign the accurate time points for the binding of ADO with AAR, as we were unable to determine the binding order of ADO and stearyl-CoA/NADPH with AAR. In addition, we cannot exclude the possibility that AAR binds ADO and stearyl-CoA/NADPH in a random order to form the ternary complex for further stabilization. Since the ternary complex formation is to facilitate the direct transfer of the aldehyde product from AAR to ADO as explained in the revised manuscript (Line 295-302), ADO may bind AAR before or after the stearyl-CoA binding, or even after the NADPH binding. Therefore, we thought there

might be several states, in which AAR is possibly binding ADO. We have added the suggestion in our revised manuscript (Line 384-391, also shown below). In addition, based on our structural and biochemical data, we have re-generated Fig. 5 (shown below), in which we have plotted the catalytic cycle of AAR-ADO complex and estimated the states of AAR associating and dissociating with ADO.

“It is noteworthy that AAR in apo state (S0) is able to associate with ADO, as shown by the ITC results in the present study as well as in an earlier report. The AAR-ADO binary complex may further bind the substrate, resulting in a stable ternary complex. Moreover, AAR alone is fully active and capable of catalyzing the conversion of acyl-ACP/acyl-CoA to aldehyde, and the formation of AAR-ADO complex is to fulfill the aldehyde transfer function. As a result AAR may form a complex with ADO at several different states, even before binding substrate (S0 state) or after binding NADPH (S3 state).”

Figure 5 in the revised manuscript. Cartoon model of the proposed catalytic process of AAR in the two-step pathway. The NTD (N), mid-domain (M) and CTD (C) of AAR are colored blue, yellow and magenta, respectively. ADO is colored grey. Note that ADO may associate with AAR at several different states (such as S1, S2 and S3), since its binding is for assisting the aldehyde transferring, but not essential for substrate/NADPH binding.

In addition, as the reviewer suggested, we have elaborated the position change of the substrate and the rotamer switch of the gate residue W178 in more detail in the revised manuscript (Line 283-289, 391-397, also shown below), and added a panel in Supplementary Fig. 13 (Supplementary Fig. 13b) to show the gate residue from two

different ADO structures.

“When we superposed the structure of PmADO (PDB code: 4PGI) on SeADO in our complex structure, we found that W178 in SeADO and R191 in PmADO are located at the same positions, with their main chain atoms and C_{β} atoms well superposed (Supplementary Fig. 13b). Our structural comparison indicated that W178 could easily change its rotamer to swing its indole ring away, thus opening the entry without inducing large movement of the main chain atoms.

.....

Following aldehyde production and complex formation, the free aldehyde product is released from AAR (S4, represented by the $AAR_{\text{stearoyl-CoA}}\text{-ADO}$ structure), which results in further conformational changes of the N-terminal region of AAR at the AAR-ADO interface. This conformational change of AAR and the moving aldehyde may lead to the altered conformation of the C-terminal residues of ADO, such as the gate residue (W178 in SeADO and R191 in PmADO), thus opening the entering tunnel in ADO.”

2. Has the author tested the enzymatic activity of the AAR-ADO complexed crystals ?

We particularly thank the reviewer for inspiring us to test the activity of AAR-ADO complex. Following the suggestion, we set up the enzymatic assay according to a previous report ¹. The reaction solution is a 100 μl system containing 20 μM SeAAR and 40 μM SeADO proteins in the buffer of 20 mM HEPES pH7.5, and mixed with 0.5 mM stearyol-CoA, 4 mM NADPH, 200 mM KCl, 0.5 mM MgCl_2 , 0.5 mM TCEP, 150 μM Ferredoxin, 50 μM FNR. The assay solution was incubated at 30 degree for five hours, and extracted using ethyl acetate (The details were provided in the method section of the revised manuscript). The extraction was analyzed through gas chromatograph mass spectrometer (GC-MS), and the amount of the products of AAR (octadecanal) and of ADO (heptadecane) was quantified according to the standard curves.

However, this method cannot be applied to AAR-ADO crystals, so we used the purified AAR protein incubated with either wild type or triple mutant (E196A/E200A/D201A) of ADO to perform the enzymatic assay. Our results showed that the wild type ADO yields more than twice of its product (heptadecane) compared with the triple mutant (E196A/E200A/D201A) of ADO (Supplementary Fig. 12 in the revised manuscript), which is unable to form the AAR-ADO complex as revealed by our binding assay. The yield of octadecanal (product of AAR) is similar in the two reaction systems. This result strongly suggests that the formation of AAR-ADO complex is to facilitate the direct transfer of aldehyde from AAR to ADO. We have added the results in our revised manuscript (Line 247-250, Supplementary Fig. 12).

Supplementary Fig. 12 in the revised manuscript. The product yield of AAR (octadecanal) and ADO (heptadecane) after incubating with each other. The wild type AAR is incubated with either wild type or triple mutant of ADO. The amount of octadecanal and heptadecane produced by wild types of AAR and ADO are normalized to 100% and shown in dark grey, while the amount of products produced by wild type and triple mutant of ADO are shown in light grey.

3. What is the gene ID for AAR and ADO?

The gene ID for AAR and ADO are 3775018 and 3775017. We have provided the information in the methods section in the revised manuscript.

4. Correction:

In fig S9, change *NpADO* to *PmADO*

Thanks for pointing out the mistake. We have corrected the label in Fig. S9 (Supplementary Fig. 13 in the revised manuscript).

Response to Reviewer 2:

Biosynthesis of alkanes in cyanobacteria has attracted great attention as a substitute for petroleum-based fuels. Cyanobacteria produces alkanes using a two-step reaction involving two enzymes, AAR and ADO. Although the crystal structures of ADO have been already known, structure of AAR has been elusive. This paper for the first time reports the structure of AAR and those of the AAR-ADO complexes in which AARs bind various ligands. The results suggest the structures of the AAR-cofactor and AAR-substrate complexes and a possible channel from AAR to ADO for efficient delivery of an aldehyde. Finally, the authors proposed a catalytic cycle involving structural changes of AAR and binding/dissociation of ADO. Although the purpose of this work is important and interesting, there are many concerns as described below.

We thank the reviewer for critically assessing our manuscript, pointing out mistakes, as well as raising important points that have helped to improve our study and strengthen our results. We greatly appreciate that the reviewer values the purpose of our work as important and interesting. The objective of this work was to report the new exciting structural and biochemical data of both AAR alone and AAR-ADO complex, which advance our understanding towards the alkane biosynthesis pathway in cyanobacteria and provide the basis for further exploration and engineering of the two enzymes.

Major concerns:

1. One of the major concerns on this paper is the direction of the acyl chain in the acyl tunnel of AAR. The present crystal structures showed that the alkyl group, instead of the aldehyde group, of the acyl chain is pointed toward the inside of ADO. If so, however, the alkyl group enters deep inside the ADO protein, where the reactive center is located, and the aldehyde group of the acyl chain cannot reach the reactive center of ADO. This is unreasonable. Thus, the present results cannot clearly explain the mechanism of efficient delivery of an aldehyde product from AAR to ADO.

Thank you for your comments, and we apologize for not describing our results clearly. We have explained it in the revised manuscript (Line 260-279 and also shown as following), and modified Fig. 5 for further elucidation. The modified Fig. 5 is also shown below.

“Previously reported crystal structures of PmADO showed that a T-shaped channel is present inside ADO, with an opening between helices 7 and 8 at the ADO surface, which was suggested to represent the entry point for aldehyde.

.....

The opening at ADO surface leads to the entering tunnel, which corresponds to one half of the horizontal arm of the T-shaped channel. The other half of the horizontal arm together with the vertical arm forms the substrate cavity, which was observed in several ADO structures solved in both the present study as well as in earlier reports

(Fig. 4c). Based on these structural observations, here we propose a model for the process of transferring aldehyde from AAR to ADO. The alkyl group enters the T-shaped channel of ADO, as a result of which the entire aldehyde molecule passes through the entering tunnel, before it reaches the distal end of the whole horizontal arm of the T-shaped channel. The aldehyde group proceeds towards the inside the horizontal arm after the transfer is completed, and upon the bending of the alkyl chain in the middle, accesses the di-iron center of ADO (Fig. 5).”

Figure 5 in the revised manuscript. Cartoon model of the proposed catalytic process of AAR in the two-step pathway. The NTD (N), mid-domain (M) and CTD (C) of AAR are colored blue, yellow and magenta, respectively. ADO is colored grey. Note that ADO may associate with AAR at several different states (such as S1, S2 and S3), since its binding is for assisting the aldehyde transferring, but not essential for substrate/NADPH binding.

We hope that we have explained the proposed delivery process of the aldehyde product from AAR to ADO more clearly in the revised version.

2. Although AAR was previously suggested to have higher activity for acyl-CoA than for acyl-ACP (ref. 15), it was later demonstrated that AAR has a strong preference for acyl-ACP substrates over acyl-CoA substrates. Since the natural substrate of AAR is acyl-ACP, the authors should use acyl-ACP rather than acyl-CoA for crystallization studies. Without the crystal structure of the complex between AAR and acyl-ACP, the mechanism of AAR catalysis cannot be solved. In this sense, the present report is preliminary and needs additional data.

It is true that acyl-ACP is preferable substrate for AAR according to the previous report ¹. However, as the reviewer pointed out and also explained in our revised manuscript (Lines 66-73), AAR is able to use both acyl-ACP and acyl-CoA as its substrate, and generate the same aldehyde product. This is because that acyl-ACP and acyl-CoA (such as Stearoyl-ACP and stearoyl-CoA) carry the same acyl moiety (stearoyl chain), which is linked to either ACP or CoA (Response Fig. 1). For example, in the stearoyl-ACP, a stearoyl chain is covalently linked with the sulphhydryl group of the 4'-phosphopanthetheine, which is covalently attached via a phosphodiester bond to the hydroxyl group of a specific serine residue of ACP (Response Fig. 1A). In the acyl-CoA, a fatty acyl chain is covalently linked with the CoA moiety, a part of CoA (the region marked with a black parenthesis in Response Fig. 1B) shows similar structure with the phosphopanthetheine moiety. The different part between acyl-ACP and acyl-CoA is the ACP and the ADP part of CoA moiety, while these parts are released from AAR after the spontaneous formation of acyl-enzyme intermediate. It is the acyl (stearoyl) chain that is covalently linked with residue C294 of AAR (Response Fig. 1C) to form the acyl-enzyme intermediate and is further reduced to stearic aldehyde during catalysis. Since the acyl-enzyme intermediate, which is undergone further reduction in the presence of NADPH, is exactly the same with both substrates (acyl-ACP and acyl-CoA), we believe it is possible that the mechanism of AAR catalysis is similar for both acyl-ACP and acyl-CoA.

Response Fig. 1 The structure of ACP protein and AAR protein. (A) Previously reported ACP structure (PDB code 5H9H). ACP has a 4'-phosphopanthetheine group (indicated by a black parenthesis mark) covalently bound with the Ser residue (indicated by a black arrow). The sulphhydryl group of 4'-phosphopanthetheine is able to covalently link with an acyl chain. (B) The structure of AAR solved in the present study. AAR protein binds one acyl-CoA molecule that is composed of an acyl chain (indicated by a blue parenthesis mark) and a CoA moiety (indicated by an orange parenthesis mark). A part of the CoA moiety (indicated by a black parenthesis mark) shows similar structure with the 4'-phosphopanthetheine group. (C) The structure of AAR solved in the present study. AAR protein binds an acyl chain (indicated by a blue parenthesis mark) which is covalently bound with Cys294 of AAR (indicated by a black arrow).

In addition, among the acyl-CoA esters of various chain lengths, AAR shows a marked preference for stearoyl-CoA as previously demonstrated ². Therefore basing on the feasibility of structural analysis, we chose stearoyl-CoA as the substrate of

AAR to perform all the experiments. The small organic molecule, such as stearyl-CoA, can be used in high concentration, which is more favorable for either co-crystallization with AAR(-ADO) or soaking with the AAR(-ADO) crystals. In comparison, stearyl-ACP is an 80-residue protein with a stearyl chain linked to the 4'-phosphopantetheine on a Ser residue, while the crystallization of protein-protein complexes (AAR-ACP) is much difficult than that of protein-ligand (AAR-stearyl-CoA) complexes.

Now following the reviewer's suggestion, we have tried to obtain the recombinant ACP protein covalently linked with stearic chain (stearyl-ACP), according to the previous literature³. However, we failed in obtaining the stearyl-ACP despite of multiple trials. Therefore we have not been able to go further in obtaining the crystal structure of AAR complexed with stearyl-ACP. Considering the similar chemical structure of the acyl part of acyl-ACP and acyl-CoA, and the time-consuming experimental processes for obtaining the stearyl-ACP protein and AAR-stearyl-ACP crystals, we would like to suggest to leave this study for the further exploration.

3. Unfortunately, the binding site of Mg²⁺ ion(s) was not determined in the present study, although this information is important in understanding the role of Mg²⁺ in AAR catalysis.

We particular thank the reviewer for this suggestion. In our originally submitted manuscript, we supposed that Mg²⁺ ion binds with AAR near its active site, but we cannot confirm this suggestion solely based on the electron density maps, as Mg²⁺ ions behave similarly as waters. Now according to the reviewer's suggestion, we have performed ITC experiment to confirm whether Mg²⁺ ions bind with AAR. Our result suggested that Mg²⁺ ions do not stably associate with AAR, at least under the present experimental conditions. We deduced that the Mg²⁺ ions might transiently bind AAR and facilitate its catalysis, but not occupy a specific position within AAR. Lacking experimental evidence, we have deleted the discussion about metal ions binding AAR in the revised manuscript.

4. The authors assumed that Mg²⁺ and K⁺ ions bind to AAR in the crystals. However, there have been no biochemical evidence showing that the metal ions do bind AAR. Therefore, it is required to confirm that Mg²⁺ and/or K⁺ really bind to AAR using ITC or other methods. Note that it is possible that K⁺ ions do not directly bind AAR but just enhance the solubility of acyl-CoA in the presence of Mg²⁺ ions. See Constantinides & Steim (1986) Arch Biochem Biophys. 250(1): 267-270.

We thank the reviewer for the important suggestion. We have now performed the ITC experiment. The result suggested that both Mg²⁺ and K⁺ do not stably/directly bind AAR under our experimental conditions. Based on these results, we agree with the reviewer that K⁺ ions may not directly bind AAR but just enhance the solubility of

acyl-CoA in the presence of Mg^{2+} ions. We have added this result and the suggested explanation in the revised manuscript (Line 201-210). Moreover, we have shortened this part and deleted the discussion about metal ions binding AAR in the revised manuscript.

5. It is also unfortunate that a long contiguous channel from AAR to ADO was not observed in the present crystal structures.

It is true that the *Se*ADO in our AAR-ADO complexes exhibits an occluded conformation, thus the long contiguous channel from AAR to ADO was not observed in our crystal structures. Up to now, there are 21 structures of (potential) ADO deposited in PDB (eight from *Prochlorococcus marinus*, *Pm*; five from *Synechococcus elongates*, *Se*; two from *Sulfurisphaera tokodaii*; one from *Synechocystis* sp. PCC 6803; one from *Gloeobacter violaceus*; one from *Oscillatoria* sp.; one from *Synechococcus* sp. CC9605; one from *Nostoc punctiforme*; and one from *Limnothrix* sp. KNUA012). In most of these structures, ADOs are in the occluded conformation with the substrate cavity occupied, only three *Pm*ADO structures (PDB codes: 4PG1, 4PGI, 4PGK) have an entering tunnel for substrate with an opening between its helices 7 and 8. The three structures were obtained by co-crystallization of *Pm*ADO with synthesized substrate analogs (a long-chain water-soluble aldehyde or a medium-chain carboxylic acid)⁴, and these analogs are located in the entering tunnel in addition to the substrate cavity of ADO, thus open the entry between its helices 7 and 8. These results suggested that the entry on ADO surface is usually shielded but is able to open under certain conditions, which may explain the reason why we did not observe the contiguous channel in our AAR-ADO complex structures.

Therefore, we superposed *Pm*ADO structure (PDB code: 4PGI) on the *Se*ADO part in our AAR-ADO complex structure to show the contiguous channel. *Pm*ADO and our *Se*ADO show high sequence identity of 63% (77% similarity), and the structure of *Pm*ADO is nearly identical to our *Se*ADO structures (Supplementary Fig. 13a in the revised manuscript). In addition, we found that although W178 in *Se*ADO structure and the corresponding residue R191 in *Pm*ADO structure show different orientation of their side chains (W178 in *Se*ADO structure covers the entry, whereas R191 in *Pm*ADO swings away thus opens the entry), these two residues are located at the same positions, with their main chain and C_{β} atoms superposed well (Supplementary Fig. 13b in the revised manuscript). This indicated that W178 could easily change the rotamer, allowing its indole ring to swing away from the entry and open it without inducing large movement of the main chain atoms of ADO. Thus we believed that the long contiguous channel observed in the superposed structural model (*Se*AAR-*Pm*ADO) is present in the *Se*AAR-*Se*ADO complex under certain conditions.

Supplementary Figure 13. in the revised manuscript. Structural comparison of SeADO and PmADO. (a) W178 in *SeADO* corresponds to R191 (PDB code 4PGI) in *PmADO*. The two residues are located at the entrance of ADO substrate tunnel and may function as a gate. *PmADO* and *SeADO* are colored yellow and cyan, respectively. (b) The main chain atoms and Cβ atoms of W178 in *SeADO* and R191 in *PmADO* can be superposed well.

6. To understand the activity of AAR and its mutants, it is necessary to measure k_{cat} and K_m values, instead of reporting just an octadecanal yield (see Table S2). Also, it is desirable to measure activities of all mutants used in the present study.

Following the reviewer's suggestion, we have performed the enzymatic assay to measure the K_m and K_{cat} values of AAR wild type and Y247F, a mutant was discussed in the manuscript. The enzymatic assay is according to the previous reports^{2,5}, by monitoring NADPH oxidation using absorbance at 340 nm to follow the reaction in real time.

However, we found that it is difficult to obtain reproducible data in the kinetic measurements, which might be due to the low activity and slow turnover of AAR as well as the product inhibition as previously suggested^{1,5}. Although after multiple trials with extensively screening for the reaction condition, using the freshly purified AAR wild type proteins, we have managed to obtain the data set to calculate the K_m and K_{cat} values, the errors are considerably large because of the poor repeatability. And we failed to obtain data of Y247F mutant, which may have much lower activity as we suggested in our manuscript, thus impeding the kinetic measurement. Therefore, we have summarized the preliminary result of the enzymatic assay in Response Fig. 2, but would like to suggest not to include the data in the revised manuscript.

Response Fig. 2 Kinetic analysis of AAR using stearoyl-CoA concentrations from 0 to 300 μM.

7. To verify the AAR-ADO interface, the authors created three ADO mutants targeting the acidic residues of helix 7 and measured the binding affinity of AAR with them using ITC. The results showed that both the E200A/D201A double mutant and the E196A/E200A/D201A triple mutant of ADO completely lost the binding affinity to AAR. However, it is still unknown which of E200 and D201 is essential for the AAR binding. In addition, contributions of E196 and E208 are unknown. To clarify these issues, it is necessary to create four additional mutants, namely the E196A, E200, D201A and E208A single mutants and study the contribution of each acidic residue of ADOO on the AAR binding using ITC. It is also recommended to make some AAR mutants at positively charged residues, to confirm the importance of electrostatic interactions between AAR and ADO.

We thank the reviewer for the important suggestion. As requested, we have created four additional single mutants of ADO, namely E196A, E200A, D201A and E208A, and measured their binding affinity with AAR through ITC. In addition, to avoid systematic error possibly induced by different batches of proteins, we also repeated the previous measurements of the binding affinity between AAR wild type and ADO (wild type and mutants E211A, E200A/D201A and E196A/E200A/D201A). Our results confirmed our previous data that the double and triple mutants (E200A/D201A and E196A/E200A/D201A) of ADO completely lose their capability to bind AAR, and showed that two acidic residues, E196 and E200, are essential for ADO in the AAR-ADO interaction. To verify the critical role of these two ADO residues, we also

created three AAR mutants (R73A, R79A and K80A), targeting the basic residues involved in the interaction with E196 and E200 of ADO, and measured their binding affinity with AAR. Consistent with our ITC data obtained with ADO mutants, the three AAR mutants all abolish the binding capability with ADO. These results strongly suggested that interaction between the basic residues (R73, R79 and K80) of AAR and acidic residues (E196 and E200) of ADO are crucial for the AAR-ADO complex formation. The results have been added to the revised manuscript (Line 126-144 and Supplementary Figs. 4, 5) and summarized in Fig. 1d.

*“To further verify the AAR-ADO interface observed in the complex structures, we mutated each of the five acidic residues from helix 7 of ADO, and further generated a double mutant E200A/D201A and triple mutant E196A/E200A/D201A (Supplementary Fig. 3). We then measured the binding affinities of AAR with the wild type and the seven mutant forms of ADO using isothermal titration calorimetry (ITC) (Fig. 1d, Supplementary Fig. 4). The binding assays revealed that the wild type forms of AAR and ADO formed a complex with a measured K_d value of 2.0~2.2 μM , which is comparable to that of the similar AAR-ADO complex from *Nostoc punctiforme* (Np). The mutants E196A and E200A exhibit significantly reduced binding affinity with AAR, whereas the E200A/D201A and E196A/E200A/D201A mutations completely abolished the capability of ADO to bind with AAR (Fig.1d, Supplementary Fig. 4).*

We further mutated the basic residues of AAR involved in the interactions with E196 and E200, creating three AAR single mutants, R73A, R79A and K80A (Supplementary Fig. 3). The binding assay between AAR (wild type and mutants) and ADO confirmed the essential role of the R73(AAR)-E196(ADO), R79(AAR)-E196(ADO) and K80(AAR)-E200(ADO) pairs for the AAR-ADO complex formation (Fig.1d, Supplementary Fig. 5). Together, our results demonstrate that the AAR-ADO complex observed in our structure is representative of the complex that exists in physiological conditions.”

8. The present results indicate that acyl-CoA and NADPH binds at the same site on AAR competitively. To verify this, it is required to confirm that acyl-CoA cannot bind to AAR simultaneously with NADPH using, for example, ITC.

We thank the reviewer for the suggestion. First we would like to mention that our suggestion about acyl-CoA and NADPH binding at the same site on AAR is based on our structural observation. In addition, this suggestion is supported by the similar chemical structure of NADPH and CoA moiety (Response Fig. 3). It is reasonable that NADPH is able to bind to the same pocket of CoA moiety after it released from AAR. Moreover, our suggestion is also in line with the previously suggested Ping-pong mechanism catalytic mechanism employed by AAR².

Response Fig. 3 Structures of stearoyl-CoA and NADPH. (A) Chemical structure (upper panel) of stearoyl-CoA and Stearoyl-CoA built in AAR-ADO complex structure (lower panel). Stearoyl-CoA is composed of a stearic moiety and a CoA moiety. (B) Chemical structure (upper panel) of NADPH and NADPH built in AAR-ADO complex structure (lower panel). The similar part in stearoyl-CoA and NADPH are circled by red dashed line.

Now following the reviewer's suggestion, we have measured the binding affinity between stearoyl-CoA and AAR incubated with NADPH by ITC method. However, the result shows that the binding between AAR and stearoyl-CoA was not significantly affected by NADPH. We assumed that the high affinity of stearoyl-CoA with AAR ($0.09 \mu\text{M}$) in the presence of K^+ ions makes it a stronger competitor that may replace NADPH within AAR. Therefore, we have turned to measure the binding affinity between NADPH and AAR incubated with stearoyl-CoA, through microscale thermophoresis (MST) method. We used both wild type and mutant C294S of AAR. The wild type AAR is able to spontaneously form the acyl-enzyme intermediate and release the CoA moiety, thus the NADPH binding pocket is vacant, whereas the mutant C294S binds the whole stearoyl-CoA, thus the NADPH binding pocket is always occupied. Therefore the C294S mutant incubated with stearoyl-CoA (C294S(-CoA)) should not be able to bind NADPH if our suggestion is correct. As expected, our MST results showed that AAR wild type is able to bind NADPH with a K_D value of $\sim 7.99 \mu\text{M}$, whereas the mutant C294S(-CoA) does not bind NADPH (Supplementary Fig. 10 in the revised manuscript). These results together confirmed our suggestion that NADPH and CoA occupy the same site within AAR. We have added the result in our revised manuscript (Line 222-228, Supplementary Fig. 10).

Name	Target Name	Ligand Name	K_D (μM)
AAR-WT	WT	NADPH	7.99
AAR-C294S	C294	NADPH	N.D.

Supplementary Figure 10 in the revised manuscript. The binding affinity of AAR with NADPH measured by microscale thermophoresis (MST). The wild type AAR (AAR-WT) binds NADPH with a K_D value of 7.99 μM . The AAR C294S mutant incubated with stearyl-CoA (AAR-C294S) is unable to bind NADPH.

9. It is required to measure the binding affinity (K_d) of ADO with the Y247A and Y247F mutants of AAR, because Y247 of AAR forms a hydrogen bond with H7 of ADO.

We apologize for not clearly describe the interaction in our original submitted manuscript. Y247 is located at the central part of AAR and is hydrogen bonded with the seventh residue His (H7) of AAR, but not involved in the interaction with helix 7 of ADO (Response Fig. 4A). To avoid confusing, we have carefully checked the revised manuscript, and make sure that we use different designation for them, i.e. H7 for residue His7 of AAR, and helix 7 for the seventh helix of ADO.

Response Fig. 4 Structures of AAR-ADO complex (A) and AAR (B). (A) AAR and ADO are colored green and cyan, respectively. The two residues Y247 and H7 in AAR are labeled. An N-terminal helix of AAR and the helix 7 of ADO are responsible for the AAR-ADO interaction. These two helices are highlighted in cartoon mode and circled by red dashed line. (B) AAR was colored slate, yellow and magenta for NTD, mid-domain and CTD. The acyl chain, together with Y247, H7 and surrounded aromatic residues are shown in sticks.

Our structures showed that Y247 is involved in the acyl-tunnel formation (Response Fig. 4B). Moreover, the hydrogen bond between Y247 and H7 as well as the hydrophobic interactions between Y247 and adjacent aromatic residues facilitate the compact folding of AAR (Response Fig. 4B), which is important for AAR to stably bind the substrate. Therefore we assumed that Y247 plays a minor role in associating with ADO, but is pivotal for substrate binding. As expected, the Y247F mutant shows decreased binding affinity with stearyl-CoA (Fig. 3c in the revised manuscript), but has similar binding affinity of 2.2 μ M with ADO (Fig. 1d in the revised manuscript) compared with the AAR wild type. However, the mutant Y247A completely abolish its binding capability with both stearyl-CoA and ADO (Figs. 1d, 3c in the revised manuscript). It is possible that the mutation of Tyr into Ala, a residue possesses a very small side chain, may induce the structural or conformational change of the whole AAR protein, leading to the complete loss of function of this mutant form. We have added this result in our revised manuscript, and modified Fig. 1 and Supplementary Fig. 5 accordingly.

Other comments:

10. In Fig. S9, W178 in SeADO is compared with R191 in PmADO. For this purpose, the authors used the PDB file of 4PGK for the structure of PmADO. However, this structure has the R191L mutation (not Val but Leu) and is not appropriate for the above purpose. The authors should use other PmADO structures that do not have mutation at R191 in Fig. S9 and Fig. 4.

Thanks for the kind indication. Following the reviewer's suggestion, we have now used another PmADO structure (PDB code: 4PGI), which possess the T-shaped

channel and does not have mutations, to compare with SeADO structure. The figures (Fig. 4 and Supplementary Fig. 13 in the revised manuscript) and figure legends were modified accordingly.

11. To solve the AAR(NADPH)-ADO structure, the authors soaked the AAR(thioester)-ADO crystal with NADPH. However, it is curious why the authors did not try to construct the AAR-ADO-NADPH ternary complex without addition of acyl-CoA.

Indeed we have tried to obtain the AAR-NADPH/AAR-ADO-NADPH complexes, by co-crystallizing AAR or AAR-ADO with NADPH. In addition, as we already obtained the AAR_{thioester}-ADO crystals, we also soaked the crystals with NADPH, and finally obtained the AAR_{NADPH}-ADO structure through the soaking method.

When we solved the AAR_{NADPH}-ADO structure, we found that the NADPH molecule is in an S-shaped conformation, similar with that in the previously reported structures^{6,7}. Its ADP moiety is accommodated by the dinucleotide recognition loop, which was previously suggested to interact with NADPH. In addition to the extensive interaction between AAR and NADPH, the hydride transferring C4 atom of NADPH and the thioacyl group of Cys294 is in van der Waals distance, which supports the functional relevance of the NADPH binding pocket showed in our structure. Based on all these results, we suppose that the NADPH binding mode observed in our structure represents its real binding manner with AAR in vivo. Therefore, we did not keep on trying in obtaining the crystals of AAR-NADPH/AAR-ADO-NADPH complexes.

12. When presenting the ITC data, the authors only showed the Kd values (see Table S2 and Figs. S3 and S6). However, to check the validity of the ITC experiments, other parameters should also be shown, including delta H, delta S, and stoichiometry.

Thank you for the important suggestion. During this revision, we have generated several more mutants of both AAR and ADO, therefore we have performed the ITC experiments again, using all the wild type and mutants of AAR and ADO. All the parameters of the ITC data were provided in the revised manuscript (Supplementary Figs. 4, 5, 8).

13. The ITC data on the binding of AAR with stearoyl-CoA, shown in Fig. S6, indicate a stoichiometry of 1:0.5, but this value is not reasonable.

Thank you for the comment. The stoichiometry of 1:0.5 in our ITC data means that only half of the AAR protein molecules bind stearoyl-CoA. We assumed that the inaccurate stoichiometry might be the result of inaccuracy of AAR quantification in our previous experiments, as a little impurity could lead to the higher estimated concentration of AAR. In addition, during this revision, we found that a portion of AAR proteins, which were pre-purified and stored at -80 °C, may not be in a state (or

conformation) that is ready to bind substrate. These may lead to the inaccurate apparent stoichiometry of 1:0.5 for AAR and stearyl-CoA.

As mentioned in the previous response (response 12), during this revision, we have performed all the ITC experiments again. By using the freshly and highly purified protein and carefully quantifying the concentration, now we have obtained more reliable ITC data. The K_D values are slightly different, but lie in a same range compared with our previous data, whereas the stoichiometry between AAR and stearyl-CoA is in a range of 1~1.2. The results have been provided in Supplementary Fig. 8 in the revised manuscript.

14. In the ITC measurements, the authors prepared an AAR sample of 720 μ M. However, since AAR is prone to aggregation (see ref. 15), there is a concern that AAR was aggregated, and the ITC measurements may not be properly carried out. If the authors have some tips to solubilize the AAR protein at high concentrations, please describe them in detail.

Thank you for the suggestion. In our experiment, we found that AAR can be concentrated to ~30 mg/ml (corresponding to 800 μ M) for the crystallization trials without precipitation. We assumed that the molecular chaperones (dnak-dnaJ-grpE-groES-groEL) co-expressed with AAR help it correctly folding and keeping it in good conditions. In addition, we had added Mg^{2+} and K^+ ions in the purification and storing buffer of AAR, which may also help to stabilize AAR proteins. The detailed expression and purification procedure have been provided in the Method section in the revised manuscript.

As the reviewer indicated, an earlier report indicated that AAR is not stable, “Purified cACR was found to be quite unstable and tended to precipitate after 3–4 days when stored at 4 °C, although it could be stored for prolonged periods at -80 °C without loss of activity.”² Based on the information, we always separated the purified AAR proteins in small volume, freeze them with liquid nitrogen, and stored the purified proteins at -80 °C. By this way, we can avoid repeatedly freezing and thawing the proteins, thus keep the proteins stable and active.

However, we would like to mention that during this revision, we have found that only the freshly purified AAR proteins exhibit the best activity, while the AAR protein stored at -80 °C is not as stable as the fresh protein.

15. The methods section should be described in more detail in Supplementary Information. To reproduce the purification of the AAR and ADO samples, it is required to describe, for example, components of media and antibiotics, timing of IPTG induction, cultivation time of E. coli, methods of determination of protein concentration and so on. In addition, please describe the amino acid sequence (22 residues) attached at the C-terminal region of AAR as a His-tag.

Thank you for your suggestion. We have described the methods in more details in the revised version.

16. The purity of the AAR and ADO proteins should be shown. For example, the data on SDS-PAGE after purification and the profiles of size exclusion chromatography using Superdex 200 should be attached in Supplementary Information.

Suggestions are well taken. We have provided the profiles of size exclusion chromatography profiles of wild types of AAR and ADO, as well as the AAR-ADO complex, and SDS-PAGE results of all the wild type and mutant proteins. The figure was added as Supplementary Fig. 3 in the revised manuscript.

17. In the AAR(apo) structure, there were three flexible regions, called Fr-I, Fr-II and Fr-III. Although roles of Fr-I and Fr-III are discussed in the manuscript, nothing is discussed on Fr-II.

We thank the reviewer for the kind indication. Fr-II is involved in the interaction with ADO. We have added this description in our revised manuscript (Line 107-110; Line 304-307).

18. Page 2, line 23-24: In the Abstract, the authors wrote that the two enzymes act in series and potentially form a complex that efficiently converts long chain fatty acids into hydrocarbons. However, this sentence is incorrect: instead of fatty acids, fatty acyl-ACP or fatty-acyl-CoA is converted into hydrocarbons.

Thanks for pointing this out. We have corrected it in the abstract.

19. The authors should clearly describe which cyanobacterial species AAR and ADO are derived from. Line 78 describes that AAR for the apo form is from Synechococcus elongatus PCC 7942. Is the same AAR used for AAR-ADO complexes? Is ADO derived from the same cyanobacterial species?

Thanks for the suggestion. Both AAR and ADO in the present study are from *Synechococcus elongatus* PCC 7942. We have clarified this in the revised manuscript (Line 84-85).

20. In Fig. S4c, the lines pointing to M21 are not properly shown.

We have re-generated Fig S4c (Supplementary Fig. 6c in the revised manuscript).

21. There are many typographical errors in the manuscript:

- Line 78: *Synechococcus elongates* 7942  *Synechococcus elongatus* PCC 7942
- Line 333: *plasmid GKJE8*  *plasmid pGKJE8*
- In Fig. 2c, *V8* is probably *L8*.
- Line 2 of Fig. S1 caption: *homologous*  *homologues*
- Line 2 of Fig. S1 caption: *elongatus* 7942  *elongatus* PCC 7942
- Line 3 of Fig. S1 caption: *Nostocacea* sp. PCC 7120  *Anabaena* sp. PCC 7120
- The title of Fig. S9: *NpADO*  *PmADO*
- Table S2: *Kd* value (*uM*)  *Kd* value (*micromM*)

Thank you. We have corrected all these mistakes and typos, and carefully checked the revised manuscript.

Reference

1. Warui, D.M. et al. Efficient delivery of long-chain fatty aldehydes from the Nostoc punctiforme acyl-acyl carrier protein reductase to its cognate aldehyde-deformylating oxygenase. *Biochemistry* **54**, 1006-15 (2015).
2. Lin, F., Das, D., Lin, X.N. & Marsh, E.N. Aldehyde-forming fatty acyl-CoA reductase from cyanobacteria: expression, purification and characterization of the recombinant enzyme. *FEBS J* **280**, 4773-81 (2013).
3. Wang, W., Liu, X. & Lu, X. Engineering cyanobacteria to improve photosynthetic production of alka(e)nes. *Biotechnol Biofuels* **6**, 69 (2013).
4. Buer, B.C., Paul, B., Das, D., Stuckey, J.A. & Marsh, E.N. Insights into substrate and metal binding from the crystal structure of cyanobacterial aldehyde deformylating oxygenase with substrate bound. *ACS Chem Biol* **9**, 2584-93 (2014).
5. Schirmer, A., Rude, M.A., Li, X., Popova, E. & del Cardayre, S.B. Microbial biosynthesis of alkanes. *Science* **329**, 559-62 (2010).
6. Faehnle, C.R., Le Coq, J., Liu, X. & Viola, R.E. Examination of key intermediates in the catalytic cycle of aspartate-beta-semialdehyde dehydrogenase from a gram-positive infectious bacteria. *J Biol Chem* **281**, 31031-40 (2006).
7. Demmer, U. et al. Structural basis for a bispecific NADP⁺ and CoA binding site in an archaeal malonyl-coenzyme A reductase. *J Biol Chem* **288**, 6363-70 (2013).

REVIEWERS' COMMENTS:

Reviewer #1 (Remarks to the Author):

This paper has a potential to be accepted, but some minor points have to be clarified or fixed.

Minor concerns

In L408 and L436, Please verify the gene ID number and is it genbank ID ?

In the legend of figure 4c, the color of ligand should be defined by the according PDB ID or protein name.

In SI figure 12, please check the title "The yield of product of AAR (octadecanal) and ADO (heptadecane) after incubating with each other."

In the legend of figure 3, footnotes for "a" and "b" are omitted

Reviewer #2 (Remarks to the Author):

All concerns raised previously are adequately answered in the revised manuscript. However, this reviewer still has some minor comments on the revised manuscript and new data as described below.

1. This reviewer is curious about the activity of the E196A/E200A/D201A triple mutant of ADO. Suppl Fig. 12 showed that the hydrocarbon yield was lower for the triple mutant compared with the wild type ADO. Two possible explanations of this result are (1) low efficiency of aldehyde transfer from AAR to the triple mutant of ADO due to the inability of AAR-ADO binding and (2) low activity of the triple mutant of ADO. Although the authors showed that the triple mutant does not bind to AAR, the second possibility still remains.
2. In line 325-327, the authors suggested that the active form (the ligand-bound form) is more favorable for AAR to bind ADO, compared with the loosely packed apo form of AAR. Are there any experimental evidence supporting this suggestion?
3. Please describe how many times the ITC and MST measurements were repeated to check the reproducibility of the data.
4. Line 184: The reference 25 cited here is not related to the activity of AAR. The reference that should be cited here is probably ref. 15.
5. Line 228: Ref. 18 should be cited here instead of ref. 25.
6. Figure 1(d): What is the meaning of "N.D."?
7. Figure 3(c): What are the meanings of superscript a and b?
8. Figure 4(c): What do the blue, yellow and red sticks show?
9. Figure 5 legend: Abbreviations of the three domains (N, M and C) are not necessary because they are not used in the Figure.
10. Suppl Fig. 3: Please describe the details on the experiments, such as molecular weights of the markers, and the column and protein concentrations used in the size exclusion chromatography.

11. Suppl Fig. 11: Orientation of the molecules in panels (a) and (c) should be the same.

12. There are many typographical errors:

Line 409: synthetized  synthesized

Line 415: knaJ  dnaJ

Line 673: measure  measured

Ref. 8: Rahmana  Rahman

Figure 1(c): R70  R73

Figures 1(d) and 3(c): "micro"  mu in Greek alphabet

Figure 3(c): ml  mL

Suppl Fig. 12, line 5: by wild type and ...  by wild type of AAR and ...

Suppl Table 1: stearyol-CoA  stearoyl-CoA

Response to the reviewers' comments

Reviewer #1 (Remarks to the Author):

This paper has a potential to be accepted, but some minor points have to be clarified or fixed.

Minor concerns

1. In L408 and L436, Please verify the gene ID number and is it genbank ID ?

Answer: Thank you for the suggestion. We have verified the gene ID number of SeAAR and SeADO, and also indicated that they correspond to the open reading frame 1594 and 1593 of the complete genome of *Synechococcus elongatus* PCC7942, respectively. The GenBank number of the genome has been provided in the revised manuscript.

2. In the legend of figure 4c, the color of ligand should be defined by the according PDB ID or protein name.

Answer: Thank you for the important suggestion. We have defined the color codes according to the protein name and PDB ID in the legend of Figure 4c.

3. In SI figure 12, please check the title "The yield of product of AAR (octadecanal) and ADO (heptadecane) after incubating with each other."

Answer: Thank you for the suggestion. We have modified the title as "Comparison of product yields of AAR incubated with either wild type or triple mutant of ADO" in the revised manuscript.

4. In the legend of figure 3, footnotes for "a" and "b" are omitted

Answer: Thank you for pointing out the mistake. We have changed Figure 3d into Table 3 according to the editorial suggestion, and added footnotes for "a" and "b" in the Table.

Reviewer #2 (Remarks to the Author):

All concerns raised previously are adequately answered in the revised manuscript. However, this reviewer still has some minor comments on the revised manuscript and new data as described below.

1. This reviewer is curious about the activity of the E196A/E200A/D201A triple mutant of ADO. Suppl Fig. 12 showed that the hydrocarbon yield was lower for the triple mutant compared with the wild type ADO.

Two possible explanations of this result are (1) low efficiency of aldehyde transfer from AAR to the triple mutant of ADO due to the inability of AAR-ADO binding and (2) low activity of the triple mutant of ADO. Although the authors showed that the triple mutant does not bind to AAR, the second possibility still remains.

Answer: We agree with the reviewer that the second possibility remains. However, it is not highly likely. Previously reported structural data of ADO alone as well as our AAR-ADO complex structure showed that the three residues are located at helix 7 of ADO, while the active site, namely the di-iron center, is located at the interior of a four-helix bundle composed of helix 1, 2, 4 and 5 of ADO. The three residues of Helix 7 are far away from the active site, with distances ranging from 16 Å to 23 Å, from the di-iron center, thus hardly affect the activity of ADO. On the contrary, helix 7 is crucial for the interaction with AAR, and the mutation of the three residues was demonstrated to lead to the incapability of ADO in binding AAR. Therefore, we suggest that the low efficiency of aldehyde transfer from AAR to the triple mutant of ADO due to the inability of AAR-ADO binding is the main reason that results in the decreased amount of heptadecane produced by ADO mutant. This suggestion is also consistent with previous study on AAR and ADO from another cyanobacterial species *Nostoc punctiforme* (*Np*), which showed that the presence of *Np*AAR led to enhanced alkane production of *Np*ADO (Warui *et al.*, *Biochemistry*, 2015, ref. 18 in the revised manuscript).

2. In line 325-327, the authors suggested that the active form (the ligand-bound form) is more favorable for AAR to bind ADO, compared with the loosely packed apo form of AAR. Are there any experimental evidence supporting this suggestion?

Answer: We indeed measured the binding affinity of AAR in both apo form and ligand-bound form with ADO during the last revision of the manuscript, and found that the ligand-bound form shows higher affinity with ADO (~ 0.07 μM) compared with the apo AAR (2.0-2.2 μM). However, the titration pattern is not perfect (please see the figure), and the error is considerably large. In addition, we were not able to repeat the measurement due to the shortage of the protein sample and the substrate stearyl-CoA at that time, so we did not show the result and only make a suggestion in our manuscript. Now we have

modified the text, replacing “our structural data showed that the compact folding of AAR in the complex is favorable for substrate/NADPH binding and ADO association” with “our structural data suggested that the compact folding of AAR in the complex is favorable for substrate/NADPH binding and ADO association” in the Discussion section for more accuracy.

3. Please describe how many times the ITC and MST measurements were repeated to check the reproducibility of the data.

Answer: The ITC measurements were repeated two to three times, the MST measurements were repeated three times, all these measurements were successfully reproduced, and the representative results were shown. The data files of ITC measurements were shown in Supplementary Figures 5, 6 and 9, and the source data file related with MST measurements (Supplementary Figure 11) have been provided along with the revised manuscript.

4. Line 184: The reference 25 cited here is not related to the activity of AAR. The reference that should be cited here is probably ref. 15.

Answer: Thank you for pointing out the mistake. We have replaced ref. 25 with ref. 15 in the revised manuscript.

5. Line 228: Ref. 18 should be cited here instead of ref. 25.

Answer: Thank you for pointing out the mistake. We have replaced ref. 25 with ref. 18 in the revised manuscript.

6. Figure 1(d): What is the meaning of "N.D."?

Answer: “N.D.” means “not detected”. We have added the explanation for “N.D.” in the legends of all related figures and tables.

7. Figure 3(c): What are the meanings of superscript a and b?

Answer: We have changed Figure 3d into Table 3 according to the editorial suggestion, and added footnotes for “a” and “b” in the Table.

8. Figure 4(c): What do the blue, yellow and red sticks show?

Answer: We have defined the color codes in the figure legend as “The *Pm*ADO structure (PDB code 4PGI) is shown in surface mode, and the ligand bound in the T-shaped channel is shown as

yellow sticks. The SeAAR-SeADO structure (colored magenta) and the free SeADO structure (PDB code 4RC5) (colored cyan) are shown in ribbon, with the bound ligand shown as sticks”.

9. Figure 5 legend: Abbreviations of the three domains (N, M and C) are not necessary because they are not used in the Figure.

Answer: Thank you for pointing out this. We have deleted the abbreviations in the legend of Figure 5.

10. Suppl Fig. 3: Please describe the details on the experiments, such as molecular weights of the markers, and the column and protein concentrations used in the size exclusion chromatography.

Answer: Thank you for the suggestion. We have provided experimental details in the Method section, and the information of the molecular weights of the markers, the column and protein concentrations used in the size exclusion chromatography in the legend of Supplementary Fig. 4 (Supplementary Fig. 3 in the originally submitted version).

11. Suppl Fig. 11: Orientation of the molecules in panels (a) and (c) should be the same.

Answer: Thank you for the suggestion. We have re-generated the figure in panel (c) to ensure the orientation of the molecules in panels (a) and (c) are the same.

12. There are many typographical errors:

Line 409: synthesized  synthesized

Line 415: knaJ  dnaJ

Line 673: measure  measured

Ref. 8: Rahmana  Rahman

Figure 1(c): R70  R73

Figures 1(d) and 3(c): "micro"  mu in Greek alphabet

Figure 3(c): ml  mL

Suppl Fig. 12, line 5: by wild type and ...  by wild type of AAR and ...

Suppl Table 1: stearyl-CoA  stearoyl-CoA

Answer: Thank you for your corrections, we have corrected all these typos and mistakes.